



# Analysis of functional groups in atmospheric aerosols by infrared spectroscopy: method development for probabilistic modeling of organic carbon and organic matter concentrations

Charlotte Bürki[1], Matteo Reggente[1], Ann M. Dillner[2], Jenny L. Hand[3], Stephanie L. Shaw[4], and Satoshi Takahama[1]

[1]ENAC/IIE Swiss Federal Institute of Technology Lausanne (EFPL), Lausanne, CH-1015, Switzerland
[2]Air Quality Research Center, University of California Davis, Davis, CA 95616, USA
[3]Cooperative Institute for Research in the Atmosphere, Colorado State University, Fort Collins, CO 80523, USA
[4]Electric Power Research Institute, Palo Alto, CA, 94304, United States

**Correspondence:** Satoshi Takahama (satoshi.takahama@epfl.ch)

**Abstract.**

The Fourier transform infrared (FTIR) spectra of fine particulate matter ($PM_{2.5}$) contain many important absorption bands relevant for characterizing organic matter (OM) and obtaining organic matter to organic carbon (OM/OC) ratios. However, extracting this information quantitatively — accounting for overlapping absorption bands and relating absorption to molar abundance — poses several challenges. For instance, a subset of model parameters lead to calibrations that test almost indistinguishably well against laboratory standards generate substantially different predictions in ambient samples. Furthermore, additional parameters related to molecular structure are required to estimate carbon content from functional group (FG) abundance. However, since many carbon atoms can be branched (not fully functionalized) or polyfunctional, these parameters are not well constrained for ambient sample mixtures.

In this work, we present a probabilistic framework to characterize combinations of these parameters that are consistent with field measurements of organic carbon (OC), for which estimates from thermal optical reflectance (TOR) measurements are used. Uncertainties in this probabilistic framework characterize the plausibility of many different parameter values that yield acceptable predictions (to the extent that they can be evaluated) neglected in conventional estimates of statistical uncertainties. Based on calibrations of aliphatic CH, alcohol COH, carboxylic acid COO, carboxylate COO, and amine NH2, we find model parameters for approximately homogeneous groups of samples determined from cluster analysis of FTIR spectra available for 17 sites in the Interagency Monitoring of Protected Visual Environments (IMPROVE) monitoring network (7 sites in 2011 and 10 additional sites in 2013). These groups are interpreted as being predominantly influenced by dust, residential wood burning, wildfire, urban sources, and biogenic aerosols.

The resulting calibrations reproduce TOR OC concentrations ($R^2 = 0.96$) and provide OM/OC values consistent with our current best estimate of ambient OC. The mean OM/OC ratios corresponding to sample types determined from cluster analysis range between 1.4 and 2.0, though ratios for individual samples exhibit a larger range. Trends in OM/OC for sites aggregated by region or year are compared with another regression approach for estimating OM/OC ratios from a mass balance of the





major chemical species contributing to PM fine mass. Differences in OM/OC estimates are observed according to estimation method and are explained through the sample types determined from spectral profiles of the PM.

## 1  Introduction

Organic mass to organic carbon (OM/OC) was originally characterized using gas chromatograph-mass spectrometry (GC-MS) data (White and Roberts, 1977; Turpin and Lim, 2001) by estimating molecular weight per carbon for individual molecules. However, the analyzed compounds only comprised a small fraction of the overall OM mass and their representativeness for actual aerosol mixtures has been a subject of perennial inquiry. An alternative approach to estimate OM from mass balance of chemical species obtained by sequential extraction has been demonstrated (El-Zanan et al., 2005; Polidori et al., 2008; El-Zanan et al., 2009), but the labor-intensive operation limits the number of samples that can be analyzed. To obtain an effective OM/OC over a large number of samples for a given site or season, regressing concentrations of a suite of particulate matter (PM) components to the gravimetric mass (via the "Reconstructed Fine Mass" equation) in monitoring network measurements has been proposed (Frank, 2006; Malm and Hand, 2007; Simon et al., 2011). However, the results can be difficult to interpret on account of combined measurement errors and intercorrelations among PM component concentrations.

In this work, we advance our ability to estimate OM/OC from Fourier Transform infrared (FTIR) spectra of PM (Allen et al., 1994; Russell, 2003; Takahama and Ruggeri, 2017). In this approach, OM and OC is estimated from organic molecular structures in the PM detected by absorption of mid-infrared radiation. The model for OC estimation from functional groups (FGs), referred to as "FG-OC", and relevant background is presented in Section 1.1. A new framework for constraining estimates through a combination of laboratory and ambient measurements and chemical simulations is described in Section 1.2.

### 1.1  OM/OC by FG estimation

Another bottom-up approach for deriving estimates of OM/OC is to use chemical measurements of atomic composition of the organic fraction using mass fragments from high resolution aerosol mass spectrometry (Aiken et al., 2008) and FGs from FTIR. Here we focus on FTIR based on its demonstrated capability to characterize $PM_{2.5}$ on Polytetrafluoroethylene (PTFE) filters collected in US monitoring networks. The original concept was outlined by Allen et al. (1994), and further developed by Russell (2003) and Ruthenburg et al. (2014).

The areal FG-OC mass density $m_C$ on each sampled filter $i$ is constructed from the areal molar densities $n$ of several FGs, denoted by the index $g$:

$$m_{C,i} = \frac{M_C}{\alpha} \sum_{g \in \mathcal{G}^*} \lambda_{C,g} n_{ig} \tag{1}$$

$M_C = 12.01$ is the atomic mass of carbon, $\alpha$ is the mass recovery fraction, and $\lambda_C$ is a coefficient that can be interpreted as the mean "fractional carbon" associated with each FG within the set of measured FGs, $\mathcal{G}^*$. Mass and molar densities typically take on units of $\mu g\,m^{-3}$ and $\mu mol\,m^{-3}$, respectively. The molar densities of each FG can be related to spectral absorbances $x$





by a separate linear model (Ruthenburg et al., 2014):

$$n_{ig} = \sum_{j \in \mathcal{J}} x_{ij} \beta_{jg}^{(k_g)} . \tag{2}$$

The basis for eq. 2 is that the absorbance due to a substance is proportional to its abundance (Beer-Lambert-Bouguer law) (Griffiths and Haseth, 2007); the coefficients $\beta$ embody the extent of overlap among target analyte and interferents, and relation between absorbance and molar densities. The coefficients are determined by calibration of laboratory standard spectra to known molar densities of FGs; however, regularization must be used to solve for $\beta$ because the number of variables (spectral absorbances) are typically greater than number of calibration samples, absorbances are multicollinear, and the inverse solution

is sensitive to small perturbations to the data. Partial least squares (PLS) regression (Wold et al., 1983; Martens and Næs, 1991) projects the spectra matrix and areal density of target analyte onto a set of common latent variables, and regularization is imposed by truncating the number of these variables. Therefore, $\beta$ is a function of the regularization parameter — the number of latent variables $k$ retained — for each FG. Further details for PLS are provided in Appendix B, and a summary of symbols related to the FG-OC model is provided in Table A1.

From the same molar densities of FGs used to estimate $m_{\mathrm{C}}$, molar densities of non-carbon atoms in set $\mathcal{A}^*$ can be added to provide an estimate of OM:

$$(\mathrm{OM})_i = m_{\mathrm{C},i} + \sum_{g \in \mathcal{G}^*} \sum_{a \in \mathcal{A}^*} M_a \lambda_{ag} n_{ig}$$

$\lambda_{ag}$ are integers relating FG abundances to composition of atoms $a$, and — unlike $\lambda_{\mathrm{C},g}$ — are well-defined. OM/OC is estimated by normalization to estimated carbon:

$$(\mathrm{OM/OC})_i = 1 + \frac{\sum_{g \in \mathcal{G}^*} \sum_{a \in \mathcal{A}^*} M_a \lambda_{ag} n_{ig}}{m_{\mathrm{C},i}} \tag{3}$$

There are two specific challenges associated with OC estimation from FGs, which also affect OM and OM/OC estimates. The first is to select the appropriate model ($\beta$) when a non-unique set of regularization parameters generate similar predictions for laboratory standards used for validation, but vary widely in their predictions in ambient samples (Reggente et al., 2019). The second is to determine a relationship between FG abundance to number of carbon atoms (through $\lambda_{\mathrm{C}}$ and $\alpha$) since many carbon

atoms can be polyfunctional, functionalized with FGs that are not measured, or not functionalized to be detectable by FTIR. The fractional carbon parameter $\lambda_{\mathrm{C}}$ take on values of unity or less to prevent multiplicitous enumeration of the same carbon atom from knowledge of FG abundance. For instance, $\lambda_{\mathrm{C,aCH}} = 0.5$ for methylene carbon leads to the correct estimate of one carbon atom for every two aliphatic CH (aCH) groups measured. Similarly, $\lambda_{\mathrm{C,aCH}} = 0.33$ corresponds to methyl carbon, $\lambda_{\mathrm{C,aCH}} = 1$ to methine carbon, and so on. Conventionally, $\lambda_{\mathrm{C}}$ was obtained by assuming the most numerous configurations of carbon

present in assumed archetypal molecules (e.g., linear hydrocarbon or ring-structured compounds). Values assumed in previous works range between 0.39 and 0.88 (Allen et al., 1994; Russell, 2003; Reff et al., 2007; Chhabra et al., 2011; Ruthenburg et al., 2014; Table S1); similar uncertainties exist for other FGs. Takahama and Ruggeri (2017) proposed an extension this approach whereby organic molecules and molecular mixtures are conceptualized as a collection of functionalized carbon atoms. Based





on the FGs for which calibrations are built, $\lambda_C$ can be estimated from the number of measured bonds on each carbon atom or by
regression over a collection of carbon atoms. Likewise, the detectable fraction of carbon atoms, $\alpha$, in molecules and molecular
mixtures can be calculated exactly within this scheme. This approach was illustrated for molecules found in the aerosol phase
from a simulation of $\alpha$-pinene photooxidation (in the presence of $NO_x$) coupled with gas/particle partitioning (Ruggeri et al.,
2016).

Parameter selection based on surrogate samples (either laboratory samples or virtual molecules in simulation) and independent estimation of ambient OC and OM is the ultimate objective for operational use of FTIR. However, there are inherent
differences in chemical composition (i.e., molecular structure, mixture complexity) between such surrogate samples or mixtures with real, ambient PM. Past studies to evaluate a limited number of parameter selection approaches, however, have led
to various degrees of agreement between FG-OC and TOR OC, and it was unclear how this bias was manifested in OM/OC
estimates reported by FTIR. Therefore, at the current stage of development, we define our objective to devise a framework to
characterize the multitude of plausible parameters that are consistent with available field measurements. Because we do not
have reference values for each FG in ambient samples, we turn to available observational data with lower chemical resolution (TOR OC) as reference, together with a probabilistic framework for providing plausible estimates for model parameters.
Despite known artifacts (Section 2), TOR OC serves as a useful target for FG-OC calibration at this stage to constrain its
parameter uncertainties; the implications of these artifacts are also taken into consideration in the model evaluation stage. This
procedure for constraining parameter uncertainties in FG-OC leads to estimation of OM/OC from FTIR that are consistent with
TOR OC.

### 1.2 Probabilistic framework

Bayes' theorem (Bayes, 1763; Robert and Casella, 2010; Gelman et al., 2013) provides the theoretical foundation for the
parameter characterization framework. Letting $p$ broadly denote any probability density or mass function, the theorem can be
written as

$$p(\theta|y) = \frac{p(y|\theta)p(\theta)}{p(y)} \tag{4}$$

where $p(y) = \int_\theta p(y|\theta)p(\theta)d\theta$. $y$ is the observed data (TOR OC), $\theta = \{\theta_1, \theta_2, \ldots \theta_D\}$ is the parameter vector of dimension $D$
(which includes unfixed FG-OC and PLS parameters), $p(\theta)$ is the prior distribution of parameters, $p(y|\theta)$ is the likelihood,
and $p(\theta|y)$ is the posterior distribution. The model for FG-OC ($m_C$, eq. 1) and explanatory variables (ambient sample spectra,
denoted by $x$ in eq. 2) corresponding to each TOR OC observation are assumed given and are excluded in this notation (Gelman
et al., 2013). In this multivariate context, a single integral denotes an integral or sum over all parameters. Notation related to
probabilistic modeling is summarized in Table A2; data and models used for each of these terms are further described in later
sections.

The inverse problem of parameter estimation in calibration is ill-posed, meaning that small differences in the input — either
data or model parameters — may lead to instabilities in the solution (i.e., parameter estimates) (Kabanikhin, 2008; Calvetti
and Somersalo, 2018). The prior $p(\theta)$ additionally provides a natural mechanism for regularization (incorporating auxiliary





knowledge) to constrain the inverse solution; with increasing number of observations effectively overriding its influence on the final estimates $p(y|\theta)$. With $\theta$ assumed to be random variables with probability distributions to be characterized according to the agreement between model and measurement, parameter uncertainty in this framework reflects the plausibility of different

models providing approximately similar predictions. As a point of contrast, conventional model fitting is formulated as solving an optimization problem to obtain a point estimate of $\theta$ that maximizes $p(y|\theta)$, with parameter uncertainties characterized by confidence intervals and model estimates by prediction intervals in the classical (or frequentist) approach to parameter estimation. Estimates of uncertainty (e.g., confidence intervals) in this case reflect the ability to characterize a single parameter value given the number of samples used and the magnitude of variations present in the data.

Bayesian inference has been used previously in atmospheric modeling (e.g., Pinder et al., 2006; San Martini et al., 2006; Thompson et al., 2011; Henderson et al., 2012; Wang et al., 2013; Tukiainen et al., 2016) for estimating under-constrained parameters using field observations in several different contexts. We adopt this approach to provide probabilistic estimates to unknown parameters; starting from prior distributions derived from laboratory measurements and available molecular structures, and updating them based on their plausibility for modeling OC as reported by TOR. In particular, the mass recovery

fraction of OC is explicitly included as an unknown parameter for estimation to allow better understanding of potentially measured and unmeasured contributions of carbon to FG-OC; separate from remaining biases with respect to the TOR measurements. We describe the measurements used in Section 2 and adaptation of this modeling framework in Section 3. Results are presented in Section 4 and concluding remarks provided in Section 5.

## 2    Experimental data

We apply this method to the Interagency Monitoring of Protected Visual Environments (IMPROVE) 2011 and 2013 data set used by Reggente et al. (2016) and Takahama et al. (2019), except that the Baengnyeong Island, South Korea, site is excluded to focus on the US sites (Figure 1). The Sac and Fox, KS, site was discontinued mid-2011 and so is not included in the analysis for the 2013 data set. Contiguous US sites are further demarcated into Northeast, Southeast, Southwest, and Northwest regions by the position 40 °N and -100 °W following the convention of Hand et al. (2019). The data set consists of reported values

and uncertainties for gravimetry, TOR, X-ray fluorescence (XRF), and ion chromatography (IC), which are used for Bayesian calibration and RCFM regression. The reported data were obtained from the Federal Land Manager Environmental Database (FED) (http://views.cira.colostate.edu/fed/; last accessed 08/16/2019).

    For functional group calibration, we use laboratory standards from Ruthenburg et al. (2014) that includes 250 samples consisting of nine type of organic compounds and organic blanks (ammonium sulfate standards with no organics). The calibration

models of Kamruzzaman et al. (2018) and Boris et al. (2019) are adapted for quantification of the amine and carboxylate content, respectively. This body of work leads to a collective set of measured functional groups $\mathcal{G}^*$ consisting of aliphatic CH (aCH), alcohol COH (aCOH), carboxylic COOH (COOH), nonacid carbonyl (naCO) (which includes ketone and ester), carboxylate COO (oxOCO), and amine NH2 (NH2). Uncertainties in PLS calibration and molecular structure parameters only associated with aCH, aCOH, and COOH are considered, since the other species did not contribute an appreciable amount to



the FG-OC over a range of parameters considered. Because of the inclusion of COOH (for which $\lambda_{C,COOH} = 1$) and additional fixed contributions from several FGs, the mass recovery parameter $\alpha$ in eq. 1 can be uniquely distinguished from $\lambda_{C,aCH}$ and $\lambda_{C,aCOH}$, leading to a model that is identifiable (Walter and Pronzato, 1997).

## 3 Statistical analysis

### 3.1 Cluster analysis of spectra

Effective model parameters for a group of samples can be estimated at the level of each site or season directly. However, estimating parameters for a group of chemically similar samples instead is favorable in that parameter values associated with molecular structure are more likely to be representative for each sample in an approximately homogeneous population. We use FTIR spectra themselves as an indicator for chemical similarity, and perform cluster analysis to create subgroups interpreted to be chemically similar. Model parameters are then applied to each member sample and aggregate statistics for OM and OM/OC are obtained for each site and seasons from their constituent samples.

Hierarchical cluster analysis (Bishop, 2009; Hastie et al., 2009) is used to categorize samples into spectroscopically similar groups (Russell et al., 2009; Liu et al., 2009; Ruthenburg et al., 2014). Spectra are first preprocessed by baseline correction (Kuzmiakova et al., 2016) and wavenumber selection (retaining only regions in the range 3700–2500 cm$^{-1}$ and 1820–1500 cm$^{-1}$) to reduce the influence of substrate interference, particle scattering, and (carbon dioxide and water) vapors in the analysis chamber (Russell et al., 2009). The spectra are then normalized by their respective L2 norms so that they vary by composition rather than absolute absorbance (which includes the effect of mass loading in addition to composition). Finally, more than 1000 wavenumbers of the normalized spectra matrix are reduced to 9 dimensions using mean-centered, unscaled principal component analysis. These 9 principal components are selected from the eigenvalue profile ("scree plot") and their capability to explain 99% of the variance of the original spectra matrix. While instrumental noise does not contribute much to the overall signal (Debus et al., 2019), this preprocessing step additionally reduces the remaining water vapor contribution to the signal that is visible in spectra with low mass loadings, and makes distance metrics used for characterizing similarity more meaningful than what can be obtained in higher dimensions of correlated variables (Domingos, 2012).

The Euclidean distance metric with complete linkage is used for clustering samples based on their principal component scores. The number of clusters are heuristically selected by examining how the overall variability is reduced within each cluster (using the within sum-of-squares metric), and how well individual samples are served by the algorithmically-determined associations (with the Silhouette coefficient) with the creation of each additional cluster (Figure S6). Eleven superclusters are selected from this procedure, and model parameters $\theta$ estimated for each cluster and applied every member within it to predict FG-OC and FG-OM. As low signal-to-noise ratio samples can adversely affect the operations involving normalized spectra (i.e., principal component and cluster analyses), 10% of samples with the lowest L2 norms are initially excluded in the procedure above, but are assigned to the most appropriate cluster through $k$-nearest neighbor ($k$-NN) classification in the principal component space a posteriori for completeness.



## 3.2 Bayesian calibration

The statistical model

$$y_i \sim \mathrm{N}(m_{\mathrm{C},i}, \sigma_i) = m_{\mathrm{C},i} + \varepsilon_i \tag{5}$$

assumes that systematic variations of TOR OC $y$ in each sample $i$ are modeled by FG-OC $m_{\mathrm{C}}$, and non-systematic contributions of measurement errors $\varepsilon$ are normally distributed with standard deviation $\sigma$ (San Martini et al., 2006; Skoog et al., 2017). The likelihood function in this model corresponds to

$$p(y|\theta) = \prod_{i \in \mathcal{S}} \left( \frac{1}{2\pi\sigma_i^2} \right)^{1/2} \exp\left[ -\frac{(y_i - m_{\mathrm{C},i})^2}{2\sigma_i^2} \right] \tag{6}$$

where the product is taken over all samples in the set denoted by $\mathcal{S}$.

Choosing a prior distribution $p(\theta)$ (eq. 4) is not a trivial task (Bishop, 2009). Where possible, it is desirable to have an informative but weak prior that does not have disproportionate impact on the results. The prior distribution also imposes bounds on the solution in that the likelihood estimated from eq. 6 are substantially downweighted in near-zero probability density regions specified by the prior (or not considered in regions where the density is identically zero for distributions with finite bounds).

We parameterize the uncertainty $\sigma^2$ in eq. 6 as

$$\sigma_i^2 = \sigma_0^2 + \kappa^2 y_i^2 \ , \tag{7}$$

with $\sigma_0^2$ denoting the irreducible error and $\kappa^2$ denoting a coefficient for the heteroscedastic (concentration-dependent) error. These terms have familiar interpretations as $2\sigma_0$ is a typical measure of the minimum detection limit (MDL), and $\kappa$ corresponds to the relative standard deviation ($\sigma/y$) in the limit of high concentrations ($y \gg \sigma_0$). $\sigma^2$ for each sample is calculated from

combined uncertainties of the thermal fractions of TOR OC, and initial estimates for these two parameters are obtained via regression of to $y$. As reported to the IMPROVE database, TOR OC uncertainties are assumed independent across samples, and correlation of errors across thermal fractions for each sample are omitted. $\sigma_0$ is kept fixed to the fitted value of $0.04 \, \mu\mathrm{g\,m}^{-3}$ as $2\sigma_0$ is higher than that reported for the TOR OC MDL ($0.05 \, \mu\mathrm{g\,m}^{-3}$) (Dillner and Takahama, 2015) and serves as a conservative estimate. The fitted $\kappa$ is approximately 7%, which is lower than collocated precision or overall errors ($\kappa \sim 15\%$) that have been

reported elsewhere (Dillner and Takahama, 2015; Brown et al., 2017). Therefore, we include $\kappa^2$ as an additional unknown parameter to be estimated, and assume a inverse gamma distribution around the fitted value for the prior. Uncertainties in $n$ and molecular structure parameters due to model variance of eq. 2 and C2 are not included in this estimate. The analytical precision (typically within 5%) is greater than that of TOR (Debus et al., 2019) but collocated precision can be similar in magnitude (Dillner and Takahama, 2015). Incorporating these considerations into eq. 6 poses additional challenges (Rock et al., 1977) and

are not considered for this study. Because of the heteroscedastic error model (eq. 7), samples with lower concentrations can have comparable or greater impact on the likelihood; the abundance of lower concentration samples (according to approximately lognormal concentrations in atmospheric samples; Ott, 1994) means a few high concentration points have less influence on parameter estimation (Section S1).

To estimate probabilities associated with the number of PLS latent variables, We use mean square error of cross validation (MSECV) typically used for model selection and convert them into probabilities using Boltzmann weighting (Appendix C1). The proposed approach leads to a prior favoring solutions with lower MSECV estimated for the calibration set (laboratory standards) and downweighting substantially high-bias (high MSECV) solutions which are not sufficiently complex to capture the spectral variations for quantification of the FG (Figure S2).

The priors for structural parameters $\lambda_{C,g}$ and $\alpha$ are estimated from virtual mixtures of primary organic aerosol compounds from automobile exhaust and wood burning measured by GC-MS (Rogge et al., 1993, 1998), and secondary organic aerosol compounds in the Master Chemical Mechanism v3.3.1 database (Jenkin et al., 1997; Saunders et al., 2003). In both data sets, compounds likely to be in the aerosol phase were selected based on volatility (equilibrium vapor concentration $C^0 \leq 10^{3.5}$ µg m$^{-3}$) (e.g., Robinson et al., 2007). Further details of the method are provided in Appendix C2 and results of analysis in Section 4.1.

Having defined the likelihood function and prior distributions, we obtain the posterior probability $p(\theta|y)$ from measurements $y$ in two ways. Our primary technique is Markov Chain Monte Carlo (MCMC), which evaluates the unnormalized posterior $p(y|\theta)p(\theta)$ for numerically sampled values of $\theta$. We also confirm our results using Laplace's method, which is a Gaussian approximation about the maximum of the unnormalized posterior. This method can only be used for continuous variables, so it is applied for each combination of $k_g$. More details on these techniques are provided in Appendix D.

## 3.3 Reconstructed fine mass regression

For comparison, we estimate OM/OC as interpreted by coefficients of the reconstructed fine mass (RCFM) equation used by IMPROVE (Malm et al., 1994; Malm and Hand, 2007; Chow et al., 2015). Given the atmospheric concentration (µg m$^{-3}$) $c$ of a substance, regression is used to obtain coefficients $a$:

$$c_{FM} - c_{EC} - c_{SS} = a_{AS}c_{AS} + a_{AN}c_{AN} + a_{OC}c_{OC} + a_{dust}c_{dust} \qquad (8)$$

*FM* is the dry fine mass concentration, measured by gravimetric analysis and corrected for particle bound water using available relative humidity measurements of the analysis laboratory and hygroscopic growth factors for constituent species as described by Hand et al. (2019). *AS* and *AN* are ammonium sulfate and nitrate, respectively, estimated from the sulfate and nitrate under the assumption of full neutralization. *SS* is sea salt, estimated as 1.8 times the chloride concentration. *dust*, also referred to as "soil," is calculated from assumed oxide forms of silicon, calcium, iron, and titanium. *OC* and *EC* are as quantified by the TOR method (Section 2). To reduce collinearity among variables, EC and SS are not included in the regression but subtracted from FM a priori (Simon et al., 2011; Hand et al., 2019). The coefficients and their confidence intervals are obtained by MLR solved by ordinary least squares (OLS) (Weisberg, 2013) and Error-in-Variables regression (EIV) (Fuller, 1987) as described by Hand et al. (2019) and Simon et al. (2011), respectively. To avoid confusion with other approaches described in this study, OLS and EIV regression for solving eq. 8 will be collectively referred to as RCFM regression. Furthermore, the results of $a_{OC}$ will be referred to as the OM/OC ratio estimate according to this approach. OLS does not consider heteroscedasticity or relative magnitude of measurement errors of any variable, which can lead to biased coefficient estimates and confidence intervals that do





not reflect the actual uncertainty (Fuller, 1987; Simon et al., 2011; Weisberg, 2013). The latter issue is addressed in this work by providing confidence intervals obtained by bootstrapping (Davison and Hinkley, 1997). EIV regression alleviates this problem by considering measurement errors of both explanatory and response variables explicitly (neglecting error covariances in this implementation); however, the estimates are subject to the accuracy of estimated measurement errors. The implementation provided by Simon et al. (2011) is used for estimation of coefficients and their uncertainties. Analytical uncertainties reported for each measurement are used for their estimates, but unaccounted systematic biases can affect the coefficient $a_{OC}$ (Hand et al., 2019).

## 4 Results

For this paper, we limit our focus on topics related to the estimation of parameters that generate FG-OC congruent with TOR OC concentrations, and comparisons of new OM/OC ratios obtained by FTIR with RCFM regression estimates. Obtaining FG composition for each filter sample enables analysis of site-specific OM/OC ratios and source-class characteristics in much greater detail, and is reserved for a separate, dedicated paper on the subject. The following subsections cover characterization of prior distributions estimated for the unknown molecular structure parameters $\lambda_\mathrm{C}$ and $\alpha$ (Section 4.1), description of spectral clusters formed (Section 4.2), posterior parameter estimates (Section 4.3), and comparison with RCFM regression (Section 4.4)

### 4.1 Prior distributions

Prior distributions of structural parameters obtained by the method described in Section 3.2 are summarized in Figure 2. The values between 0.46–0.48 for $\lambda_\mathrm{C,aCH}$ are consistent methylene (CH2) group structures, though another reason this narrow distribution can occur is that single aliphatic CH bonds are often found together with one other measured FG on the same carbon atom (Takahama and Ruggeri, 2017). In such cases, a value of $\lambda_\mathrm{C,aCH}$ close to 0.5 prevents double counting of carbon by the two bonds (Section 1.1). The broad values for $\lambda_\mathrm{C,aCOH}$ reflect the diverse carbon types in which alcohol groups are found. The $\alpha$ value centered around 0.74 reflects the undetected carbon fraction, typically missed due to branched molecular structure or functionalization by unmeasured FGs.

Several examples for molecules with incomplete carbon recovery ($\alpha < 1$) are shown in Figure 3. More generally, the types of carbon atoms undetected vary widely in their structure (Figure S4). These molecules contain unfunctionalized carbon atoms (only bonded to other carbon atoms) and carbon atoms functionalized by, for example, aldehyde, peroxide, aromatic, phenolic, and organonitrate groups, which have absorption bands in the mid-infrared but are not included in our set of calibrations. These FGs have not been prioritized for calibration following the hypothesis that molecules with these functionalities are not found in great abundance in IMPROVE samples. Aldehydes are susceptible to hydration in aqueous solutions, leading to formation of alcohols (Schwarzenbach et al., 2002; Takahama et al., 2013b). Peroxides have been shown to be labile under various (light and dark) conditions (Epstein et al., 2014; Krapf et al., 2016). Phenolic OH and aromatic groups exhibit sharp absorption peaks near 3500 and 3100 $\mathrm{cm}^{-1}$, respectively (Bahadur et al., 2010), which are not observed in ambient sample spectra; in





previous studies, Russell et al. (2011) suggested the aromatic and unsaturated FGs contributed to less than 5% of OM mass. Organonitrates also hydrolize in the presence of water to form alcohols and nitric acid (Liu et al., 2012; Zare et al., 2019), and organosulfate FGs are not included in this analysis but their contribution to the overall OM mass concentration is often bound to be less than a few percent (Hawkins et al., 2010; Russell et al., 2011; Takahama et al., 2013a; Budisulistiorini et al., 2015; Hettiyadura et al., 2017). Additionally, oxygen has been found to be the heteroatom contributing most to the variability OM/OC ratios in ambient samples (Pang et al., 2006).

The procedure of parameter updating with ambient OC estimates can help place these values in the proper context. Previous estimates of FG-OM generally reported agreement of 70–100% for submicron OM compared against AMS (Russell et al., 2009; Gilardoni et al., 2009; Corrigan et al., 2013), and FG-OC was estimated to be 60–70% of TOR OC in PM$_{2.5}$ in the IMPROVE network samples (2011 data set) (Ruthenburg et al., 2014; Reggente et al., 2019). While these differences have been partially attributed to incomplete mass recovery of carbon by FTIR, now the estimated mass recovery fraction based on molecular structure information is included explicitly into the calibration model.

In reporting OM/OC using eq. 3, we can expect a systematic underestimation of OM/OC on account of unmeasured FGs. An alternative estimate can be obtained by considering the OM/OC of only the measured, functionalized carbon (i.e., using $\alpha m_C$ for normalization in eq. 3). This latter approach can on average lead to a more representative value of the overall OM/OC (Figure S5) in oxygenated aerosol. For this work, we use eq. 3 which likely provides a lower bound on OM/OC and a means to gauge improvement in OM/OC estimates with the inclusion of additional FG calibrations.

## 4.2 Cluster descriptions

While the primary objective of cluster analysis for this study is to create chemically homogeneous groups for parameter estimation, we include a brief remark on interpreted source classes or composition associated with each spectra type. For this analysis, we use spectral characteristics visualized in Figure 4, concentrations of tracer species or magnitude of tracer variables (Figure S7; consisting of RCFM components and additional trace elements analyzed by XRF), and location and time of occurrence as indicators of source classes (Figure S8).

Clusters 1 and 4 are high sulfate, low organic samples found predominantly in rural areas; suggesting the likely association of the organic fraction with biogenic secondary organic aerosol (SOA); samples in cluster 1 are found predominantly in the SE and NE with a notable absence in the Southwest. Nearly half of samples in Clusters 2 and 5 are found in urban areas — particularly in Phoenix, AZ — and the remaining found in rural areas are likely influenced by nearby urban sources. Clusters 3, 8, and 11 occur predominantly in the Southwest and are associated with mineral dust as evidenced by sharp Si-O-H peaks above 3500 (Reggente et al., 2019), and supported by observations of elevated contributions of elements: Al, Ca, Fe, Si, and Ti. Clusters 6 and 7 occur predominantly in the Southeast and largely consist of samples originally identified by Ruthenburg et al. (2014) as being "anomalous" in their agreement of FG-OC with respect to TOR OC. Reggente et al. (2019) later proposed that these samples contained large ammonium sulfate and ammonium nitrate particles (consistent with IC concentrations) that exhibited an optical artifact known as the Christiansen peak effect, which leads to an increase in transmittance in the vicinity of the wavelength where i) the refractive index of the substance approaches that of air and ii) the particle size and



wavelength of radiation also become similar ($\sim$3300 cm$^{-1}$). Thus, these samples share a particular absorbance profile and quantification based on assumption of Beer-Lambert law can be challenged in some wavenumber regions — especially near

the absorption band of alcohol aCOH — for these samples. Samples in clusters 9 and 10 are associated with burning. For purposes of interpretation, cluster 9 is split according to child nodes of the hierarchical clustering tree into wildfire (cluster 9a) and residential wood burning (cluster 9b) groups, which are labeled according to their occurrence during a known fire period (Rim) and during winter months where residential burning takes place (Phoenix, AZ) (more in Section 4.3).

Previous work in cluster analysis with aerosol FTIR spectra resolved differences among urban (fossil fuel combustion),

terrestrial vegetation (burning and non-burning), and marine aerosols (e.g., Russell et al., 2009; Liu et al., 2009; Takahama et al., 2011; Corrigan et al., 2013). These studies focused on spectra collected during short, intensive field campaigns (typically considering samples from a single location and single season each) with higher time resolution (typically four hours), and used an inlet with nominal size cut of one micrometer. Spectra types from monitoring networks are not expected to have a direct correspondence to their work due to the use of a 2.5 micrometer size cut (more influence of dust and larger inorganic particles)

and time resolution (24 hours) of measurements (more mixing of source classes and degrees of aging). In particular the naCO fraction in IMPROVE network samples have been estimated to be negligible using several methods (Reggente et al., 2019), while naCO varies substantially across spectra types in the submicron samples collected during intensive field campaigns and have been used as indicators of biogenic and biomass burning aerosol (Russell et al., 2011). Nevertheless, some similar spectra categories are found through differences apparent in absorption profiles.

That such a large number of samples from a wide range of sites and seasons are considered together in this work suggests that selecting a limited number of clusters for statistical estimation is likely to provide only a crude separation in chemical and spectral variations that differentiate source classes or mixture proportions of source classes. In addition, first differentiation in spectra (i.e., initial branches of the hierarchical tree) is determined by ammonium NH, alcohol aCOH, and carboxylic COH, as their broad absorption bands comprise a substantial portion of the absorbance in the spectrum. These factors can lead to

clusters which contain both rural and urban samples that differ primarily by aliphatic CH absorption (which affects the overall OM/OC but not its oxygenated fractionation), and surprising associations across regions (e.g., Fresno, CA, samples associated with samples in the SE in the same cluster). However, for the purposes of parameter estimation this level of disaggregation is found to be computationally tractable and sufficient in that estimates for smaller subsets of spectra do not substantially change the OM/OC estimated with this limited number of clusters.

## 340  4.3  Estimated parameters

Estimates of parameter distributions obtained by MCMC are generally confirmed by the Laplace method (Figure 7 shown as an example for a single cluster and Figure S9 for all clusters). Therefore, the following results will focus on results of MCMC analysis. The posterior distributions for most parameters show a departure from the mode of their prior distributions, suggesting that the results are not dominated by influence of the priors. The mode of each posterior parameter distribution

for every cluster is shown in Table 1. The number of latent variables $k_{\mathrm{aCH}}$ and $k_{\mathrm{aCOH}}$ vary by cluster, suggesting that a different model is appropriate for different spectra types (and presumably different types of PM). The mass recovery fraction





$\alpha$ ranges between 0.57 and 0.83 consistent with the range estimated for primary and secondary OM species (Section 3.2). Given our expectations for low abundance of unmeasured FGs (Section 4.1), low $\alpha$ may indicate a surprising amount of branched molecules with unfunctionalized carbon atoms — though we cannot rule out the need to examine additional FGs or

that some systematic discrepancies (e.g., in absorption coefficients) between molecules in laboratory and ambient samples are also incorporated into parameter estimates. $\lambda_{aCH}$ is consistently near 0.48, at the exception of cluster 3, but possibly due to the strong prior. $\lambda_{aCOH}$ varies much more substantially across clusters and this is likely due to the different configurations of the carbon atom functionalized by aCOH. The coefficient $\kappa$ for heteroscedastic measurement error varies between 0.13 and 0.31, which is greater than the reported TOR OC analytical error of 0.07. The variations in $\kappa$ across clusters may partially

reflect differences in thermal fractions or sensitivity to different types of compounds, but is more likely reflects the range of discrepancies between modeled and measured OC across samples that arises from a given set of parameter values. Nonetheless, the estimates of remaining parameters are robust with respect to this assumption, as assessed with simulations in which $\kappa$ is kept fixed at the prior estimate of 0.07.

The comparison of fitted FG-OC with reference TOR OC (Figure 5) with 95% intervals of the posterior predictive distri-

bution (Robert, 2007; Vehtari and Ojanen, 2012; Gelman et al., 2013; Section S4) shows reasonable agreement ($R^2 = 0.96$). There is an underprediction for several high concentration samples due to the larger number of samples with lower concentrations that collectively influence the likelihood (Section S1). Posterior predictive distributions are symmetric, and FG-OC estimated from their modes are almost identical to that obtained from single-point estimates of parameters obtained as the mode of their respective distributions (Figure S11). TOR OC measurements are out-of-range of 95% prediction intervals of the

posterior distribution approximately 5% of samples. No abnormalities are detected in spectra upon investigation, which may indicate that these samples are not well-served by the current calibration model (e.g., the selection of calibration standards). That the cluster containing anomalous samples (clusters 6 and 7) can reproduce TOR OC — in contrast to previous works of Ruthenburg et al. (2014) and Reggente et al. (2019) — is surprising, but that the alcohol aCOH is estimated to be zero can be due to the effect of anomalous dispersion (Section 4.2) and some compensation may be incorporated into the value of $\alpha$ for

these samples.

Figure 6 shows the mean OM and OM/OC for each spectra type. Trends in OM estimates across these types are consistent with trends in TOR OC, with burning samples (clusters 9 and 10) exhibiting the highest OM and biogenic and dust-related samples (clusters 1, 3, 4, 8, and 11) having the lowest OM, on average. Samples with urban influences (clusters 2 and 5) have, on average, lower OM/OC than those more associated with oxidized, biogenic (clusters 1 and 4). The high alcohol aCOH

contribution to OM/OC in the dust samples (clusters 3, 8, and 11) may be indicative of condensed secondary OM (Murphy et al., 2006; Hawkins et al., 2010; Takahama et al., 2010), but may also partially be due to misappropriated hydroxyl groups or hydrates of water associated with inorganic substances (Hudson et al., 2008; Frossard and Russell, 2012). Wildfire burning samples (cluster 9a) consistently display higher OM/OC than residential wood burning samples (cluster 9b). Because these two sample types occur during warm and cold months, respectively, the contribution of photochemical aging relative to emission

characteristics cannot be easily determined from this type of analysis.





Some variability in OM/OC across samples are present within several clusters. For instance, cluster 9 of the eleven original clusters exhibited a bimodal distribution in OM/OC from distinguishable contributions from urban wood burning and rural wildfire samples (Figure S10, and have already been disaggregated for discussion (Section 4.2). Within clusters 1, 2, and 5, contrast in OM/OC ratios between samples from urban and rural sites can be observed, with values lower by ∼0.2 in the former.

Further inspection of child nodes do not clearly separate urban and rural samples as with cluster 9, and this is largely because urban and rural samples in the same cluster differ primarily by the aliphatic aCH content while the oxygenated groups are present in similar proportions. Due to its sharp peaks, aCH absorbances comprises a small portion of the overall variation in spectra considered in the clustering technique and does not exhibit substantial influence in cluster determination. The OM/OC distribution samples in clusters containing dust-influenced samples are broad (regardless of site type) due to the high variability

in estimated alcohol aCOH content.

## 4.4 Spatial and temporal characteristics

A large number of samples are required to evaluate meaningful difference in coefficients due to the number of RCFM components, range of variations in their concentrations, and their combined measurement errors. Therefore, multiple sites or multiple years of data for a given site are often used for analysis Simon et al. (2011); Hand et al. (2019). For this work, we report coeffi-

cients for the combined years of 2011 and 2013 and sites aggregated by region (restricted to those for which FTIR spectra are available, Section 2) to examine spatial and seasonal differences, or six sites for which FTIR spectra are available both years to examine temporal trends between the two years.

Estimates across regions and seasons for the two years combined are shown in Figure 8. Given the limited number of sites analyzed in this work, the region labels are used only to summarize results across multiple sites and may not be indicative of

results for the entire region. For instance, the highest OM/OC estimated by OLS for all (∼160) IMPROVE sites between 2011 and 2015 were found in the Southeast and Northeast regions (Hand et al., 2019), whereas their annual average values are, on average, below that of the Northwest region according to the sites and years considered in this study.

Estimated trends in OM/OC between EIV and OLS are consistent in that they generally predict higher OM/OC during spring and summer, except in the Northwest sites where the highest OM/OC is observed in the winter. This type of agreement is not

unexpected as the two methods use the same mass balance approach and concentration measurements. However, OM/OC estimates from OLS (ranging between 1.4–2.5) generally underestimates that from EIV (1.5–3.1) by ∼0.3 on average. This pattern of underestimation was also reported previously (Simon et al., 2011) — this difference may be partly due to the disproportionate impact of high OC (and low OM/OC) samples on squared residuals and subsequent regression coefficient estimates by OLS, which are downweighted by uncertainties in EIV that increase together with concentration. The large

confidence intervals for the Northwest and Northeast sites reflect the fact that only one or two sites are included in these regions, and displays the limit of resolution by the RCFM regression approach for limited sample sizes. Smaller confidence intervals shown for FTIR estimates reflect the fact that regional estimates are calculated as the mean of OM/OC values obtained for each sample. Magnitude of uncertainties in FTIR OM/OC due to posterior parameter uncertainties (Hoff, 2009) for any individual sample is typically below 6%, but can be higher for samples in two clusters (Section S4).





FTIR estimates of OM/OC for these regions (1.7–2.2) are on average more similar to OLS than EIV but show less variability across regions and seasons. In general, we expect that FTIR estimates reported here may be conservative (low) if important FGs are missing in our calibration models (Section 4.1). While mean OM/OC ratios and FG composition can be estimated for each location or period explicitly, its magnitude can be roughly anticipated by i) the frequency of cluster types (Figure S13) and ii) variability of OM/OC within each cluster (i.e., urban samples having lower OM/OC in each cluster; Section 4.3).

Disaggregating FTIR estimates by site type reveals that seasonal differences are greater in urban areas (∼0.2 between winter and summer) while less pronounced in rural areas (Figure 9); regional averages are more indicative of trends in the latter because there are fewer urban sites and hence smaller number of samples. OM/OC distributions indicate that rural samples over all seasons and urban samples during the summer have a mode close to 1.8, which is the assumed OM/OC multiplier currently assumed for the IMPROVE network. Phoenix, AZ, is an urban site that exhibits particularly extreme differences

in OM/OC, with low values due to wood burning and possibly less aged urban emissions in the winter (cluster 9b and 5, respectively), and high values from the influence of dust particles in the spring and summer (clusters 5 and 8) (Figure S13). The broad OM/OC distribution during these warmer months is due to the variability in alcohol aCOH contribution estimated for the dust-influenced samples. More generally, the higher OM/OC ratios estimated for the Southwest sites — particularly HOOV (Hoover, CA), BLIS (Bliss, CA), and MEVE (Mesa Verde, CO) — during the spring season are due to the prevalence

of dust-impacted samples. Because organic mass loadings of these dust-impacted samples are relatively low (Section 4.2), the mean OM/OC values during spring are similar to that of summer months if ratios are alternatively calculated taking OC-weighting into account. The higher OM/OC estimated during the spring (1.93) in comparison to summer (1.76) in the single Northeast site (Proctor Maple, VT) is not confirmed by the other two methods as their seasonal differences are not statistically significant, but inspection of spectra types indicates that the biogenic-type samples (cluster 4) were prevalent during the spring

while more urban-influenced samples (cluster 5) with lower OM/OC values were found during the summer in comparison.

       Considering only the six sites — Phoenix, AZ; Olympic, WA; Proctor Maple, VT; St. Marks, FL; Mesa Verde, CO; and Trapper Creek, AL — for which FTIR measurements are available between 2011 and 2013, we compare differences in mean OM/OC ratios (Figure 10). Hand et al. (2019) previously reported increasing trends in mean OM/OC ratios between 2011 and 2013 over the entire network; particularly with an increase of ∼0.2 during summer months. OLS and EIV for these sites also

show increasing OM/OC (by 0.35 and 0.5, respectively) for the summer months for the subset of sites analyzed in this work, and a difference of 0.4 is also significant for OLS for the spring months. However, FTIR estimates show no such trend, and the FG composition is also remarkably consistent between the two years at these sites (Figure 11). The sample type composition determined by the FTIR spectra between the two years are also similar (Figure S14), which explains this similar estimate of OM/OC. Inspection of other regression coefficients of eq. 8 indicate other changes such as decrease in $a_{dust}$ between the

two years, which may suggest changing atmospheric composition or changes in analytical bias (Hand et al., 2019) that affect estimates of $a_{OC}$. This comparison reinforces the need for further evaluation to interpret $a_{OC}$ from RCFM regression as a surrogate for the OM/OC ratio (Hand et al., 2019).



## 5    Conclusions

We presented a new framework to enable estimation of OM and OM/OC from FG calibrations of FTIR spectra that are also

consistent with the current best estimate of ambient OC, which is taken from TOR measurements. In contrast to RCFM regression approaches that estimate OM/OC from from mass balance of all other major components contributing to particulate fine mass, estimation of this metric by FTIR uses spectra of particles collected on PTFE filters together with laboratory standards of organic molecules. In contrast to standard multivariable optimization approaches for parameter estimation, the proposed probabilistic approach incorporates prior knowledge of model parameters based on performance against laboratory standards

and sensible structural parameter values derived from atmospherically-relevant molecules compiled from measurements or computer models. While this information was exclusively used for parameter determination in previous works, the Bayesian framework used here weighs plausibility of parameter values against ambient observations. The clustering approach used for selecting subgroups with similar spectral profiles also leads to estimation of model parameters that better reflect samples in each subgroup, and provides a way for associating model parameters and OM/OC estimates to various chemical classes of PM.

Model parameters that reproduce TOR OC measurements could be found for more than 94% of samples; this approach also identifies samples for which calibration models are potentially unsuitable. Spectra types associated with dust, wildfire, residential wood burning, urban, and biogenic-influenced samples were found in the IMPROVE 2011 and 2013 samples. Mean OM/OC ratios for various locations or periods are consistent with occurrences of these spectra types. In contrast to RCFM regression methods, no consistent increase in OM/OC was found between 2011 and 2013, and the spectra type composition

was also consistent between the two years.

  This work enables many directions for future studies. OM/OC ratios and FG composition can be further related to sources and specific sites or seasons for the samples introduced in this calibration study. Furthermore, the framework is described generally such that it can be applied to samples in monitoring networks or chamber experiments, and systematically evaluate improvements in calibrations with new standards or FGs. Parameters that can be applied to new samples for prediction can

potentially be determined by assessing spectral similarity of new samples to the sample types established through cluster analysis. For increasingly refined spectral types, hierarchical Bayesian modeling (Gelman and Hill, 2007) can be used to model relationships among subgroups (e.g., spectral clusters) overcome limitations in dealing with smaller sample sizes, albeit with added complexity. Additional constraints — such as residual FM (Boris et al., 2019) or comparison to additional measurements of FGs (Decesari et al., 2007; Ranney and Ziemann, 2016) — can be introduced to the maximum likelihood expression to

explore solutions which are consistent with other available measurements.

## Appendix A:  Notation

Table A1 describes mathematical symbols for carbon estimation model and Table A2 for Bayesian modeling.





## Appendix B: Partial least squares calibration

The origin of the regularization term in eq. 2 specifically for PLS regression is explained in this section. The nonlinear iterative
least squares (NIPALS) algorithm (Wold et al., 1983) is used to project a matrix of mean-centered laboratory standard spectra
with absorption $x_{ij}$, defined for each wavenumber $j$ (indexed from 1 to $J$) and sample $i$, onto a basis set of spectral profiles
(loadings) whose elements are $p_{\ell j}$, with $\ell$ representing the index of the reduced dimension (also referred to as a latent variable
or component). The PLS scores $t_{i\ell}$ embody both the contribution of component $\ell$ to the spectra and its contribution to the FG
abundance (determined by gravimetric analysis for known aerosol composition) after additional scaling by coefficient $q_{\ell g}$:

$$n_{ig}(k_g) = \sum_{j=1}^{J} x_{ij}\beta_{jg}^{(k_g)} + e_{ig} = \sum_{\ell=1}^{k_g} t_{i\ell}q_{\ell g} + e_{ig}$$

485                                                        $\forall g \in \mathcal{G}^*$                                                        (B1)

$$x_{ij}(k_g) = \sum_{\ell=1}^{k_g} t_{i\ell}p_{\ell j} + e_{x,ij}$$

For a selected value of $k_g$, the components beyond $k_g+1$ comprise the residual terms $e_{x,ij}$ and $e_{ig}$. Using the provided training
samples, $q$, and $p$ are found such that the new variables $t$ maximize the covariance with $n$ during the calibration process. Each
new spectrum (of laboratory and ambient samples) are then projected onto this basis set and its scores used to estimate the FG
abundance.

## Appendix C: Estimation of priors


### C1   Number of latent variables $k$

For each FG, we estimate a prior for the number of latent variables (denoted as $k$ rather than $k_g$ in this section for readability) by
Boltzmann weighting (Adamson, 1979) of their mean squared error of cross validation (MSECV) from laboratory calibrations.
The MSECV is written in terms of the chi-square statistic $\chi^2$:

$$p(k) = \frac{\exp\left(-\chi_k^2/2\right)}{\sum_{k=1}^{K}\exp\left(-\chi_k^2/2\right)} \text{ where } \chi_k^2 = \frac{N \cdot MSECV_k}{s^2} \ . \tag{C1}$$

$s^2$ is the expected magnitude of error, which we use as a scaling variable fixed to the condition that $\chi^2/(N-k-1) = 1$ (reduced
chi-square is unity) for the minimum MSECV solution. The form of eq. C1 is also consistent with the notion of likelihood ratios
used in model selection and Akaike weighting (Burnham and Anderson, 2003). The upper limit on $k$ is selected to balance
inclusiveness of plausible solutions against computational considerations; for each component $k$ is chosen to include several
solutions within one standard error of the MSECV and exclude physically unrealistic ones (with high proportion of negative
predictions in concentration). The choice of upper limit for $k$ can change the overall probability, but the relative probability
among solutions remain approximately similar for a range of upper limits considered.



## C2 Carbon fractions $\lambda_C$ and mass recovery fraction $\alpha$

This work extends the approach of Takahama and Ruggeri (2017) to study functionalization at the level of each carbon atom

for a larger set of atmospherically-relevant molecules with known structure. We consider the set of molecules in primary aerosols $\mathcal{M}_{\text{primary}}$ from GC-MS measurements by Rogge and co-workers (Rogge et al., 1993, 1998) previously analyzed for FG composition by Ruggeri and Takahama (2016); and the set of gas-phase photooxidation products $\mathcal{M}_{\text{secondary}}$ from MCM v3.3.1. Considering species with equilibrium vapor concentrations $C^0 \leq 10^{3.5}$ µg m$^3$, there are 193 molecules in $\mathcal{M}_{\text{primary}}$ and 1221 molecules in $\mathcal{M}_{\text{secondary}}$ (Figure S3).

A subset of molecules $\mathcal{M}^{(s)}$ are constructed by varying the fraction $\zeta$ of primary vs. secondary aerosol molecules between 0 and 1 by 0.05 increments, and randomly sampling from the required number from each population to satisfy the balance:

$$|\mathcal{M}^{(s)}| = \zeta^{(s)}|\mathcal{M}^{(s)}_{\text{primary}}| + (1 - \zeta^{(s)})|\mathcal{M}^{(s)}_{\text{secondary}}|$$

where $|\cdot|$ denotes the cardinality (number of elements) of the set. To accommodate the limited number of primary compounds available for random selection, the total number of molecules $|\mathcal{M}^{(s)}|$ considered for any subset was 50–150 so that each

contained a random subset of $\mathcal{M}_{\text{primary}}$ even for $\zeta^{(s)} = 1$. We therefore estimate $\lambda_C$ by nonnegative least squares regression of measurable carbon abundance on FG abundances repeated over various subsets $s$:

$$n^*_{C,i} = \sum_{g \in \mathcal{G}^*} \lambda^{(s)}_{C,g} n_{ig} + e_i \quad \text{where} \quad n^*_{C,i} = \sum_{k \in \mathcal{C}^*} n_{C,ik} \quad \forall i \in \mathcal{M}^{(s)} \tag{C2}$$

$n_{C,ik}$ is the number of carbon atoms for molecule $i$ in carbon type $k$, which is summed over detectable carbon types $\mathcal{C}^*$. $n_{ig}$ is the number of FGs $g$ in molecule $i$ for the measured set $\mathcal{G}^*$. The carbon associated with carboxylic COOH is subtracted from

$n^*_{C,i}$ before regression since $\lambda_{C,\text{COOH}} \equiv 1$, and only aliphatic CH and alcohol aCOH is included in the fitting procedure. The detectable carbon fraction is estimated from the same mixtures by normalizing the abundance of detectable carbon over the total carbon (denoted by set $\mathcal{C}$):

$$\alpha^{(s)} = \left( \sum_{i \in \mathcal{M}^{(s)}} \sum_{k \in \mathcal{C}^*} n_{C,ik} \right) \Big/ \left( \sum_{i \in \mathcal{M}^{(s)}} \sum_{k \in \mathcal{C}} n_{C,ik} \right).$$

$p(\lambda_{C,g})$ and $p(\alpha)$ are derived from the distribution of values estimated over realizations of subsets $s$.

## Appendix D: Sampling the posterior distribution


Eq. 4 is typically posed as a mathematical problem to obtain the posterior distribution, written in this Section as $\pi(\theta) = p(\theta|y)$ for simplicity, from its unnormalized estimate $\tilde{\pi}(\theta) = p(y|\theta)p(\theta)$:

$$\pi(\theta) = \frac{1}{Z}\tilde{\pi}(\theta) = \frac{1}{Z}e^{-L(\theta)}. \tag{D1}$$

$L(\theta) = -\log \tilde{\pi}(\theta)$ is referred to as the loss function and $Z$ is the normalizing constant (integral of $\tilde{\pi}(\theta)$ or $e^{-L(\theta)}$). In our model

(eq. 1), we have both discrete and continuous parameters which we discriminate with superscripts $(d)$ and $(c)$, respectively. To





explicitly expound on this notation, $\theta^{(c)} = \{\alpha, \kappa^2, \lambda_{C,g} : g \in \mathcal{G}^*\}$, $\theta^{(d)} = \{k_g : g \in \mathcal{G}^*\}$, and $\theta = \theta^{(c)} \cup \theta^{(d)}$. With $\theta'_i = \theta \setminus \{\theta_i\}$ denoting the set of all parameters except $\theta_i$ (i.e. the complement of $\theta_i$ with respect to $\theta$), the marginal posterior distribution for $\theta_i$ is given by

$$\pi(\theta_i) = \frac{1}{Z} \sum_{\theta_i'^{(d)}} \int_{\theta_i'^{(c)}} \tilde{\pi}(\theta_i, \theta_i'^{(d)}, \theta_i'^{(c)}) d\theta_i'^{(c)} \,, \tag{D2}$$

with $\tilde{\pi}(\theta_i, \theta_i'^{(d)}, \theta_i'^{(c)}) = p(y|\theta_i, \theta_i'^{(d)}, \theta_i'^{(c)}) p(\theta_i, \theta_i'^{(d)}, \theta_i'^{(c)})$. As with integral notation in eq. 4, the single integral or summation symbol applies over all parameters in the indexed set: i.e., $\int_\theta = \int_{\theta_1} \int_{\theta_2} \cdots \int_{\theta_D} \cdot d\theta_1 d\theta_2 \ldots d\theta_D$ and $\sum_{\theta_i'^{(d)}} = \sum_{\theta_{i,1}'^{(d)}} \sum_{\theta_{i,2}'^{(d)}} \cdots \sum_{\theta_{i,D(d)}'^{(d)}}$. A summary of notation for posterior sampling is provided in Table A2. We use Markov Chain Monte Carlo (MCMC) as our primary tool to sample $\pi(\theta)$. To diagnose convergence and accuracy of the MCMC calculations, we additionally use a simple approximation (Laplace method) to confirm our parameter distributions. We first summarize

Laplace method as it is a close extension of maximum likelihood estimation (MLE) typically used in conventional parameter estimation before describing MCMC sampling.

### D1 Laplace method

The Laplace approximation (Tierney and Kadane, 1986; Murphy et al., 2012) solves eq. D1 and D2 by making a local Gaussian approximation to the posterior distribution of the continuous variables about their maximum a posteriori (MAP) estimate (i.e.,

maximum of the function $\tilde{\pi}$). This method improves on the classical MLE approach through the weighting of a prior (for a flat prior, the MAP estimate is equivalent to the MLE estimate), and estimating probabilities from the surface curvature of eq. D1 in the vicinity of the MAP. The approximation only applies in the domain of continuous parameters, so the calculation is performed for every selected realization of discrete parameter combinations. The probability estimate is formulated from the normalization constant of a multivariate normal distribution, with $D^{(c)} \times D^{(c)}$ Hessian $H_{\theta^{(c)*}}$ of $L$ about $\theta^{(c)*}$:

$$\pi(\theta^{(c)}, \theta^{(d)}) = \left[\frac{\det H_{\theta^{(c)*}}}{(2\texttt{pi})^{D^{(c)}}}\right]^{1/2} e^{-\left[L(\theta^{(c)}, \theta^{(d)}) - L(\theta^{(c)*}, \theta^{(d)})\right]} \quad \forall \theta^{(d)} \,. \tag{D3}$$

Laplace's method is typically associated with a second-order Taylor series expansion about $\theta^{(c)*}$ which further provides the approximation: $L(\theta^{(c)}, \theta^{(d)}) - L(\theta^{(c)*}, \theta^{(d)}) \approx \frac{1}{2}(\theta^{(c)} - \theta^{(c)*})^T H_{\theta^{(c)*}}(\theta^{(c)} - \theta^{(c)*})$ for each realization of $\theta^{(d)}$. Covariance among the continuous variables can further be obtained from the inverse of the Hessian matrix. The marginal posterior for each realization of the variable $\theta_i$ is obtained by a Gaussian approximation for each integral in eq. D2 and calculating the

$D^{(c)} - 1 \times D^{(c)} - 1$ Hessian $H_{\theta_i'^{(c)*}}$ about the MAP defined as $\theta_i'^{(c)*} = \arg\max_{\theta_i'^{(c)}} \tilde{\pi}(\theta_i, \theta_i'^{(c)}, \theta_i'^{(d)})$:

$$\pi(\theta_i) = \sum_{\theta_i'^{(d)}} \left[\frac{\det H_{\theta^{(c)*}}}{(2\texttt{pi})\det H_{\theta_i'^{(c)*}}}\right]^{1/2} e^{-\left[L(\theta_i, \theta_i'^{(c)*}, \theta_i'^{(d)}) - L(\theta^{(c)*}, \theta^{(d)})\right]} \,. \tag{D4}$$

While analytically elegant and deterministic, the Laplace approximation is best suited for applications that primarily involve real (continuous) variables with a single mode in its probability density, or in the limit of large $N$ as the density converges to a normal one (Bernstein-von Mises Theorem). However, its Gaussian estimates can become unreliable toward domain boundaries





that might be imposed due to physical constraints, or in the limit of large number of variables when the high-dimensional space tends to become non-Gaussian.

We screen solutions by finding the MAP for each combination of discrete parameter values using L-BFGS-B, and removing those which are $10^{20}$ less than the absolute maximum. $\theta^{(c)*}$ for each realization of $\theta^{(d)}$ is found using L-BFGS-B, a box-constrained, limited-memory extension of the quasi-Newton method BFGS. BFGS uses an approximation of the Hessian matrix to steer its search. The Hessian matrix is not recomputed at each iteration but updated using the secant equation to account for the curvature estimated during the most recent step (Nocedal and Wright, 2006). While L-BFGS-B provides simultaneously provides estimation of the Hessian matrix with the MAP, as it is based on an approximation for the purposes of speeding up the optimization, we recompute these matrices and their determinants from numerical differentiation at the corresponding MAPs.

## D2   MCMC

MCMC (Bishop, 2009; Aster et al., 2013) approximates the posterior probability $\pi(\theta)$ from an algorithmically-generated Markov sequence $\{\theta^{[1]}, \theta^{[2]}, \dots, \theta^{[t]}, \dots, \theta^{[n]}\}$. This sequence or chain is constructed through a series of trial and acceptance moves. The Metropolis-Hastings algorithm (Metropolis et al., 1953; Hastings, 1970) describes conditions under which the generated sequence fulfills the conditions of detailed balance necessary for convergence toward a stationary (statistically invariant) distribution. For any $\theta^{[t]}$, a candidate value $\theta^*$ is generated from a proposal distribution $q(\theta^*|\theta^{[t]})$. $\theta^*$ is designated as the next value in the sequence $\theta^{[t+1]}$ with acceptance probability $a(\theta^{[t]}, \theta^*)$, defined to preserve detailed balance for a move from $\theta^{[t]}$ to $\theta^*$:

$$a(\theta^{[t]}, \theta^*) = \min\left\{1, \frac{q(\theta^{[t]}|\theta^*)\tilde{\pi}(\theta^*)}{q(\theta^*|\theta^{[t]})\tilde{\pi}(\theta^{[t]})}\right\} . \tag{D5}$$

The ratio $\tilde{\pi}(\theta^*)/\tilde{\pi}(\theta^{[t]})$ has been used in place of $\pi(\theta^*)/\pi(\theta^{[t]})$ so that explicit evaluation of the normalization constant $Z$ (eq. D1) is not required. For a symmetric proposal distribution, $q(\theta^{[t]}|\theta^*) = q(\theta^*|\theta^{[t]})$ and further simplification to eq. D5 can be obtained (Metropolis algorithm). Assignment of $\theta^{[t+1]}$ is implemented by comparison of $a(\theta^{[t]}, \theta^*)$ against the realization $u$ of a random variable uniformly distributed over $[0,1]$:

$$\theta^{[t+1]} = \begin{cases} \theta^* & \text{if } a(\theta^{[t]}, \theta^*) > u \text{ and} \\ \theta^{[t]} & \text{otherwise.} \end{cases}$$

The initial value $\theta^{[0]}$ of the Metropolis-Hasting algorithm is set at the maximum a posterior (MAP) estimated for the Laplace method. Proposal distributions for the discrete parameters $k_g$ are truncated normal distributions which bounds the range of possible values. For continuous variables, the covariance matrix $\Sigma$ of the target distribution is estimated using the first iterations of sampling, after which efficient proposal distributions are defined (Gelman et al., 2013):

$$q(\theta^{[t]}|\theta^*) \sim N(\theta^*, c^2\Sigma) \quad \text{where} \quad c^2 \approx 2.4/\sqrt{D} .$$

Two MCMC chains were run for each model, and convergence was monitored using chain trace plots and Gelman-Rubin diagnostics (Gelman and Rubin, 1992). The posterior probability distribution $p(\theta)$, marginal distributions $p(\theta_i)$, population

statistics of $\theta$ (including covariances), and posterior predictive distributions (Section 3.2) are then calculated from the numerically sampled sequence.

The distribution-free approach of this technique makes it applicable to discontinuous, non-differentiable functions, solutions at constraint boundaries, and to smaller datasets where the limiting distribution need not be normal. Sampling across models for model selection can also be handled by a special case of Metropolis-Hastings — transdimensional or reversible jump MCMC

— in which the number of parameters for each model can vary (Green, 1995; Gallagher et al., 2009). While candidate PLS solutions generated with a different $k_g$ (eq. B1) can also be interpreted as different models, for this study, $k_g$ is treated as a discrete tuning parameter for the PLS model corresponding to a fixed calibration set. The typical downside of MCMC is the high computational cost, as large number of samples are needed for convergence and to ensure that the parameters sampled non-independently can provide adequate characterization of the target density. Where possible, use of MCMC together with

simpler methods to confirm results is recommended (Brooks et al., 2011).

*Author contributions.* ST, AMD, and SLS conceived of the project. CB wrote the code, performed simulations, and analyzed results. MR prepared calibration models and guidance on their use. ST and CB wrote the manuscript; AMD and JLH provided regular input on the analysis and further editing of the manuscript. ST provided overall supervision of the project.

*Code availability.* Code for posterior sampling by MCMC and Laplace approximation is available at https://gitlab.com/aprl/fgoc-bayes.

*Competing interests.* The authors declare no competing interests.

*Acknowledgements.* The authors would like to thank Prof. Anthony Davison for helpful suggestions regarding Bayesian statistics and the Electric Power Research Institute contract 10003745 for funding.



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



845 **Figures**

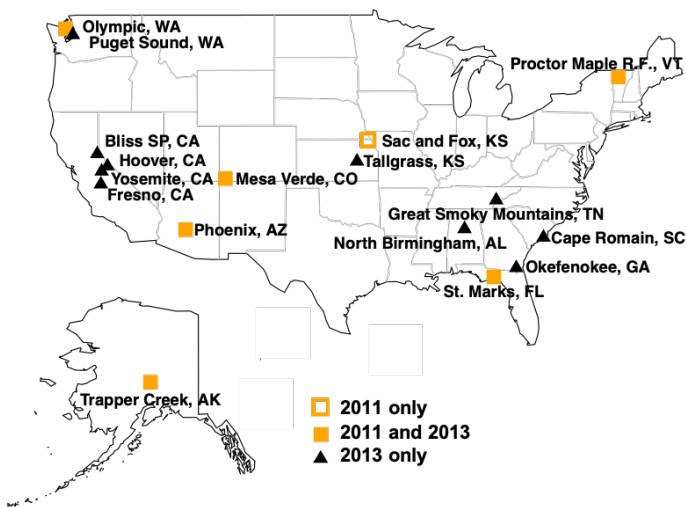

**Figure 1.** Map of IMPROVE network monitoring sites used in this work.

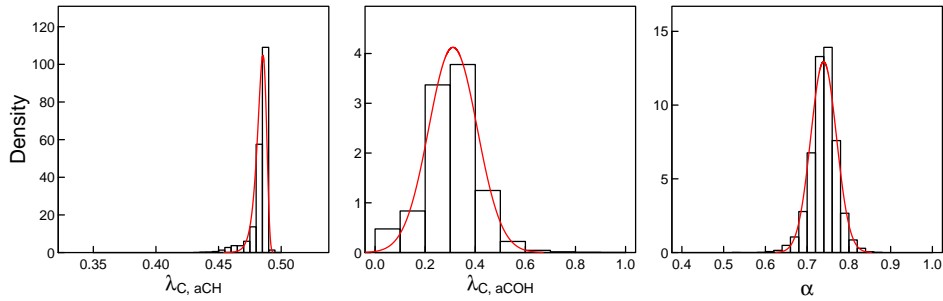

**Figure 2.** Prior distributions for $\lambda_C$ and $\alpha$. Histograms are generated from estimates from subsets of molecules representing a combination of primary and secondary organic aerosols, and red lines are fitted parametric distributions (Weibull for $\lambda_C$ and normal for $\alpha$).





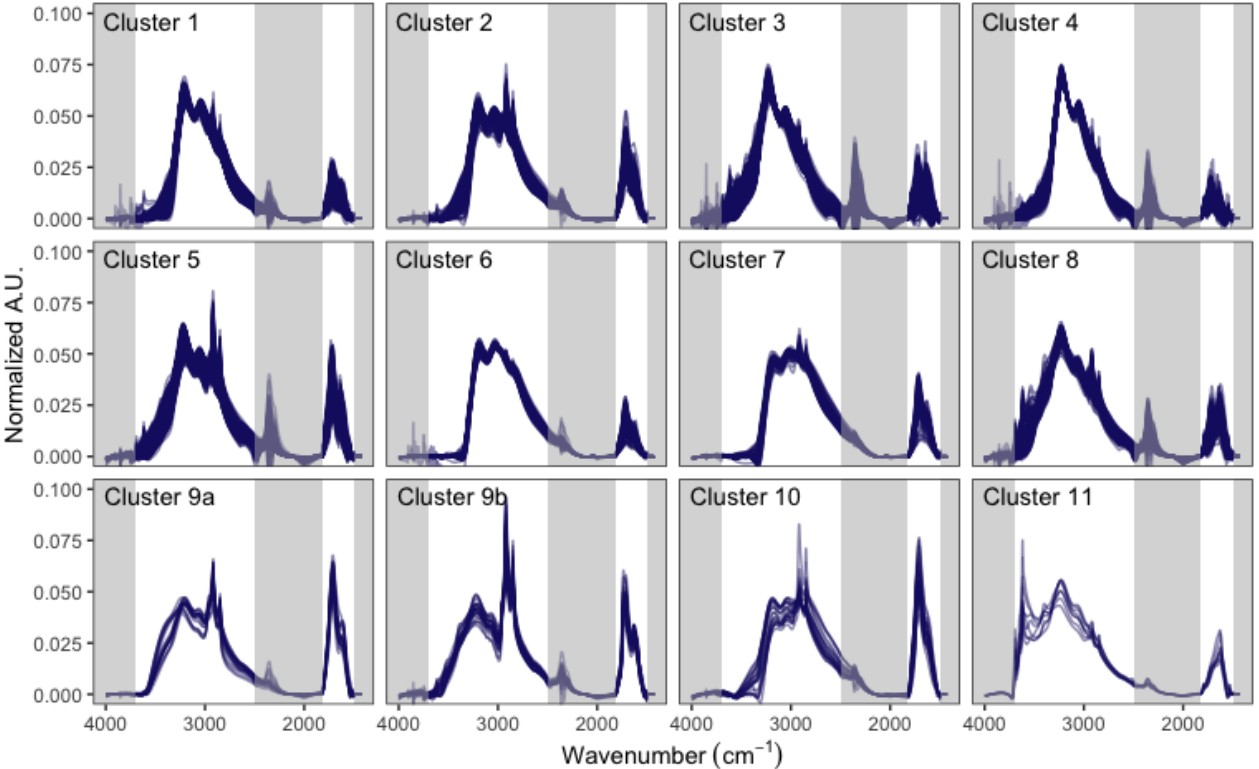

**Figure 3.** Molecules containing carbon not detected by measured set of FGs. "C721CHO" and "C1010NO3" are names designated in the MCM v3.3.1 mechanism.

**Figure 4.** Visualization of spectral clusters. Gray vertical bars indicate regions excluded from cluster analysis.





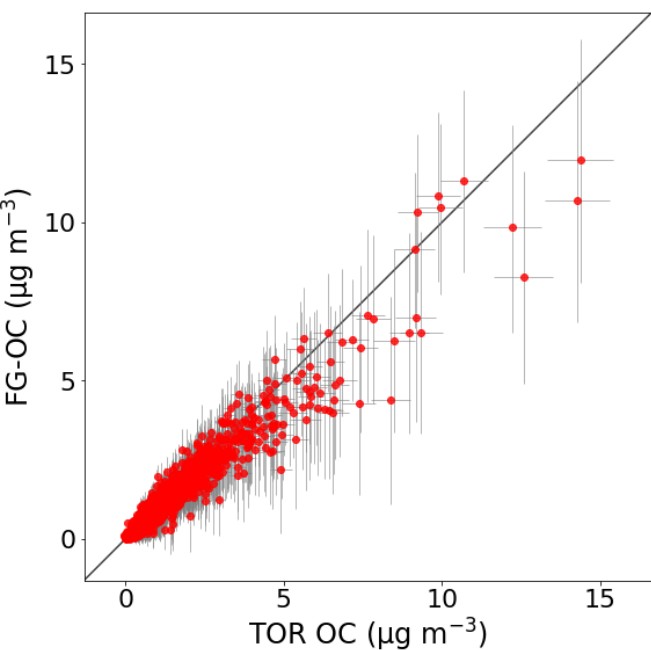

**Figure 5.** Comparison of reference TOR OC measurements and FG-OC estimated by Bayesian calibration. FG-OC corresponds to the mode of the posterior predictive distribution $\tilde{y}$ (Section S4). The lines span the 95% uncertainty intervals in TOR measurements horizontally, and 95% prediction intervals of the posterior distribution vertically. Diagonal line corresponds to 1:1 relation for reference.





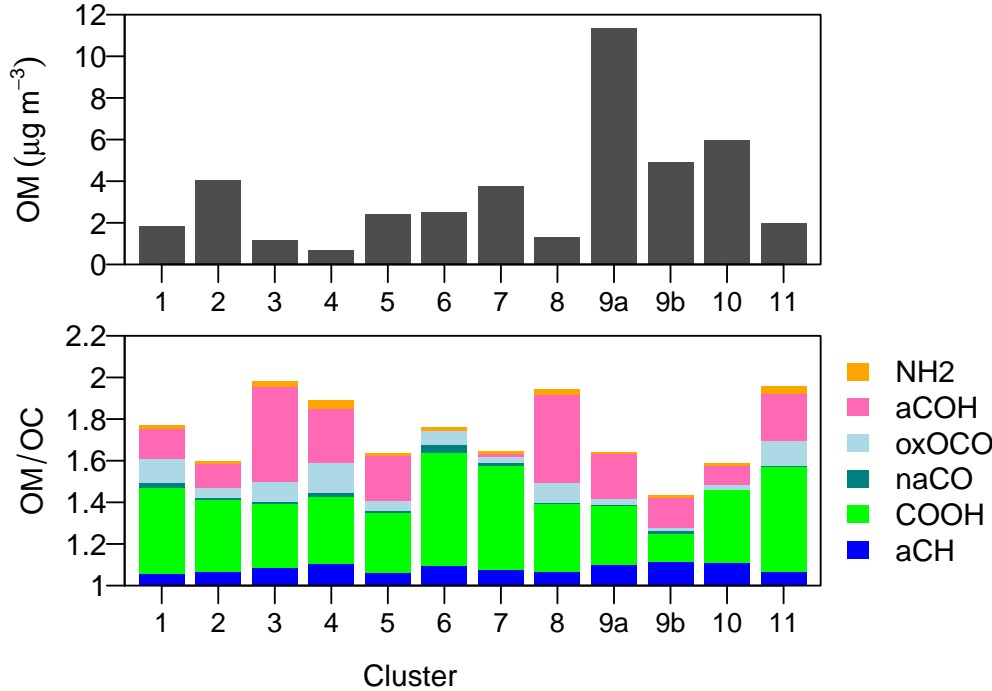

**Figure 6.** Mean OM and OM/OC for each cluster. Colors indicate FG contributions to the OM/OC.

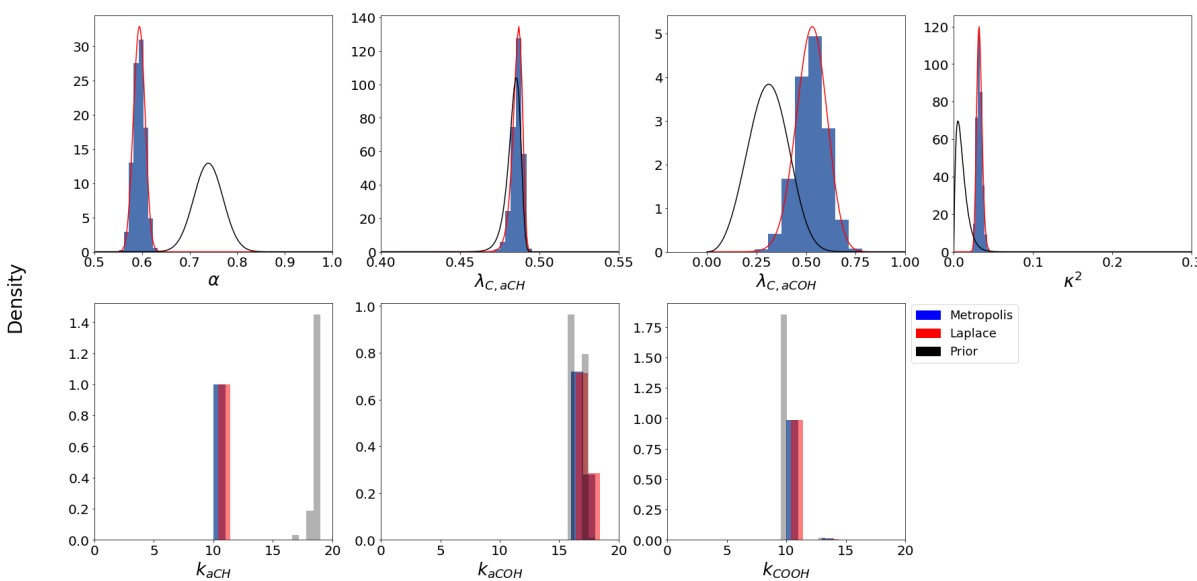

**Figure 7.** Example posterior distribution of cluster 2 from MCMC. Dark lines correspond to prior distributions, blue histograms correspond to sampled posterior distributions, and red lines correspond to Laplace estimation




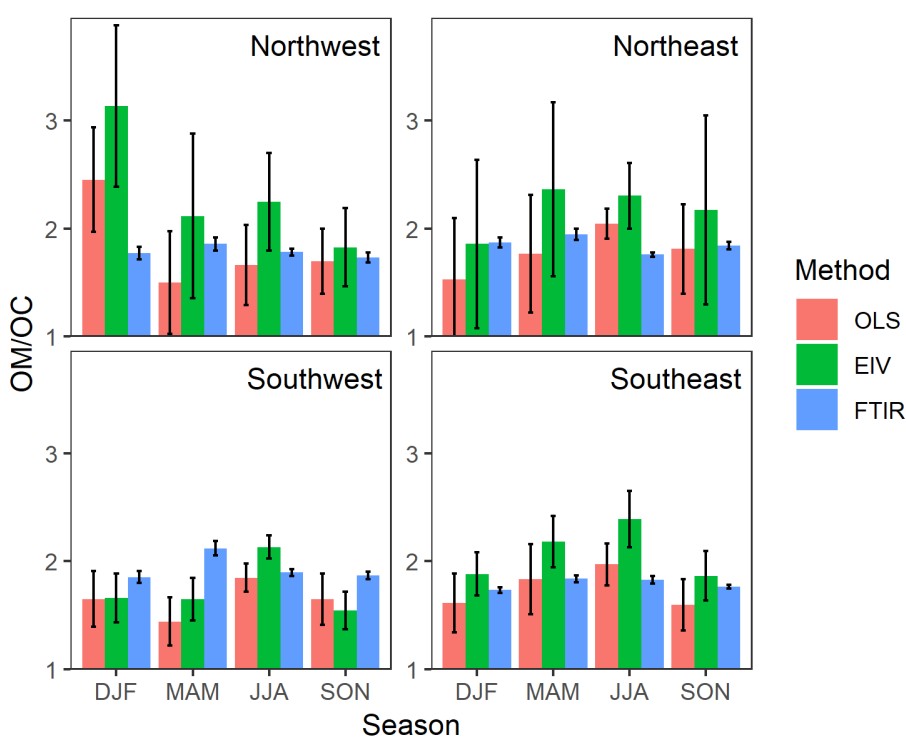

**Figure 8.** Estimates of OM/OC with 95 % confidence interval made by different techniques for the same sites for which FTIR measurements are available (Section 2). OLS (ordinary least squares) and EIV (error-in-variables) provide solutions to RCFM regression, and FTIR estimates are constructed from contributing functional groups. X-axes denote seasons: DJF = winter, MAM = spring, JJA = summer, and SON = fall.





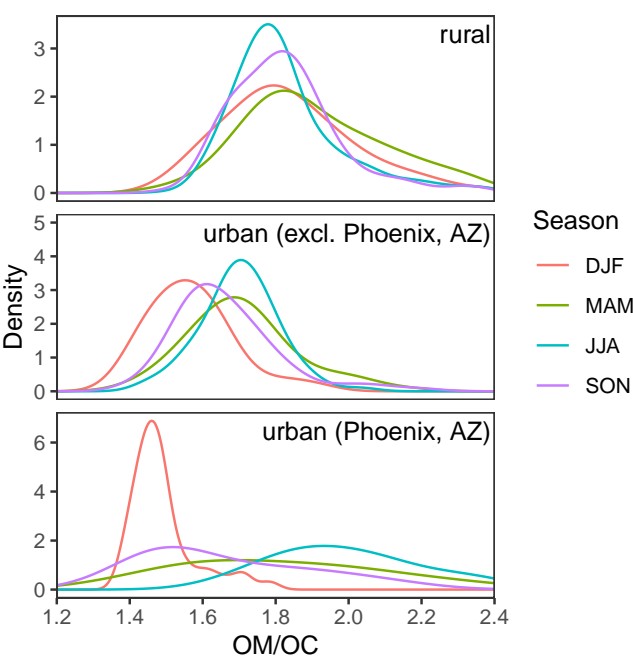

**Figure 9.** Probability densities of OM/OC estimated by FTIR for sites included in Figure 8, separated by site type. Densities for urban sites are separated into Phoenix, AZ, which is shown in its own panel, and the remaining five sites.

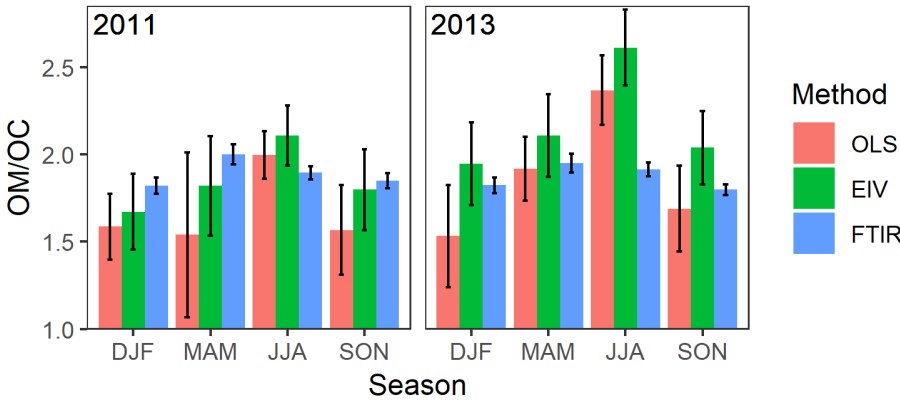

**Figure 10.** Estimates of OM/OC with 95 % confidence intervals for the same six sites for which FTIR measurements are available (one urban and six rural sites). The same notation as Figure 8 is used.





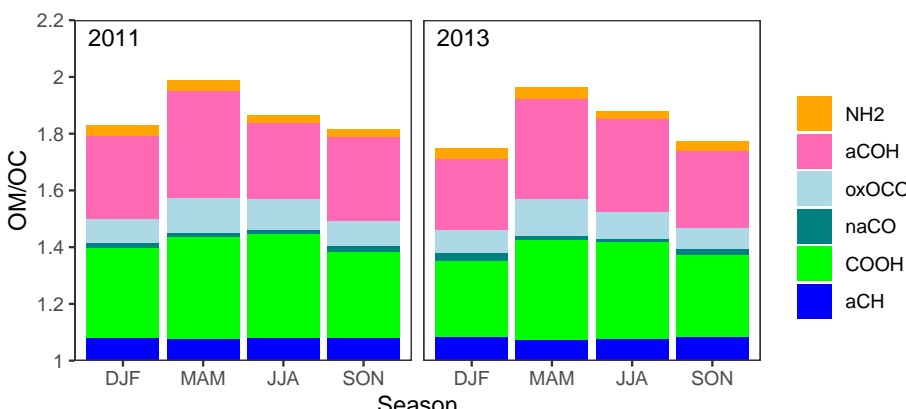

**Figure 11.** Mean OM/OC ratios partitioned by FG contributions for the FTIR estimates shown in Figure 10.





## Tables

**Table 1.** Mode of parameter posterior distributions for each cluster.

| Cluster | # samples | $\alpha$ | $\lambda_{aCH}$ | $\lambda_{aCOH}$ | $k_{aCH}$ | $k_{aCOH}$ | $k_{COOH}$ | $\kappa$ |
|---------|-----------|----------|-----------------|------------------|-----------|------------|------------|----------|
| 1 | 387 | 0.59 | 0.49 | 0.44 | 7 | 16 | 10 | 0.20 |
| 2 | 176 | 0.60 | 0.49 | 0.53 | 10 | 16 | 10 | 0.18 |
| 3 | 771 | 0.81 | 0.43 | 0.59 | 13 | 12 | 10 | 0.27 |
| 4 | 442 | 0.83 | 0.48 | 0.07 | 16 | 17 | 10 | 0.31 |
| 5 | 343 | 0.57 | 0.49 | 0.44 | 10 | 16 | 10 | 0.17 |
| 6 | 87 | 0.80 | 0.48 | 0.58 | 10 | 16 | 10 | 0.20 |
| 7 | 68 | 0.66 | 0.49 | 0.59 | 10 | 16 | 10 | 0.20 |
| 8 | 128 | 0.71 | 0.48 | 0.50 | 10 | 16 | 10 | 0.29 |
| 9 | 43 | 0.76 | 0.48 | 0.37 | 16 | 16 | 10 | 0.13 |
| 10 | 21 | 0.79 | 0.48 | 0.21 | 16 | 16 | 10 | 0.13 |
| 11 | 8 | 0.71 | 0.49 | 0.32 | 9 | 16 | 10 | 0.17 |





**Table A1.** Notation for carbon estimation model.

| Symbol | Description |
| --- | --- |
| $n$ | moles (in areal density) of atom or functional group |
| $x$ | infrared absorbance |
| $\lambda$ | number of atoms per functional groups |
| $\alpha$ | carbon mass recovery fraction |
| $m$ | mass of atom |
| $M$ | atomic mass |
| $t$ | PLS scores |
| $p$ | PLS $X$-loadings |
| $q$ | PLS $Y$-loadings |
| $e$ | model residuals |
| $k$ | number of latent variables in PLS model |
| $\mathcal{G}^*$ | set of functional groups that are measured |
| $\mathcal{A}^*$ | set of non-carbon atom types that are measured by $\mathcal{G}^*$ |
| $\mathcal{C}$ | set of carbon types |
| $\mathcal{C}^*$ | set of carbon types that are measured by $\mathcal{G}^*$ |
| $n^*$ | moles (in areal density) of a unit measured by $\mathcal{G}^*$ |
| $\mathcal{M}$ | set of molecules |
| $|\mathcal{M}|$ | number of molecules in set |
| $\zeta$ | fraction of primary to total (primary and secondary) |





**Table A2.** Notation for Bayes theorem, likelihood, and posterior sampling algorithms.

| Symbol | Description | Definition |
|--------|-------------|------------|
| $y$ | data (observations); also outcome variable | TOR OC |
| $\theta$ | set of all parameters | $\theta^{(c)} \cup \theta^{(d)}$ |
| $\theta^{(c)}$ | set of continuous parameters | $\{\alpha, \lambda_{C,aCH}, \lambda_{C,aCOH}, \kappa^2\}$ |
| $\theta^{(d)}$ | set of discrete parameters | $\{k_{aCH}, k_{aCOH}, k_{COOH}\}$ |
| $\theta'_i$ | set of continuous parameters that excludes $\theta_i$ | $\theta \setminus \{\theta_i\} = \{\theta'^{(c)}_i, \theta'^{(d)}_i\}$ |
| $\theta'^{(c)}_i$ | set of continuous parameters that excludes $\theta_i$ | $\theta^{(c)} \setminus \{\theta_i\}$ |
| $\theta'^{(d)}_i$ | set of discrete parameters that excludes $\theta_i$ | $\theta^{(d)} \setminus \{\theta_i\}$ |
| $D$ | number of dimensions (parameters) | |
| $p$ | probability density or mass function | |
| $\pi, \tilde{\pi}$ | normalized and unnormalized posterior | |
| $L$ | loss function | $\log \tilde{\pi}$ |
| $Z$ | normalizing constant | |
| $H$ | Hessian matrix | |
| $q$ | proposal distribution | |
| $a$ | acceptance probability | |