# Peer review of "Analysis of functional groups in atmospheric aerosols by infrared spectroscopy: method development for probabilistic modeling of organic carbon and organic matter concentrations"

_Atmospheric Measurement Techniques, 2019_

## Referee Comment (RC1) · Anonymous Referee #1 · 10 Dec 2019

The manuscript describes a statistical model and a probabilistic framework to characterize combinations of organic matter, organic carbon and functional groups obtained from the Fourier transform infrared (FTIR) spectra of fine particulate matter (PM2.5). The model was found to be consistent with field measurements of organic carbon (OC). The Development of these models and frameworks is important and timely as they can be used in developing machine learning algorithms.

[Figure]

The manuscript is publishable after addressing the following minor comments:

The abstract is too long. Consider shortening.

The introduction is missing a discussion on the results obtained from NMR spectroscopy in the characterizing organic aerosols and using multivariate analysis to correlate functional groups with sources of aerosols. For example, see Chapter Two - NMR Studies of Organic Aerosols, Annual Reports on NMR Spectroscopy, Volume 92, 2017, Pages 83-135.

Figure 4: this figure has 12 sub figures; each is labeled with a 'cluster' number and contain a number of overlapping spectra as judged by the gradation in the line colors. The caption needs to be modified to refer the reader to appropriate table or graph containing the definition of each cluster.

Figure 7: The caption needs to be modified to refer the reader to appropriate table of graph containing the definition of parameters in the y- and x-axes.

Figure 8,9,10,11: the use of three-letter abbreviation for seasons that do not correspond to the actual name of the season is confusing: DJF = winter, MAM = spring, JJA = summer, and SON = fall. It is better to use WR = winter, SG = spring, SR = summer, and FL = fall.

---

## Referee Comment (RC2) · Anonymous Referee #2 · 28 Dec 2019

The manuscript describes a novel approach using FT-IR and functional groups to obtain organic matter to organic carbon ratios. The manuscript fits the scope of Atmospheric Measurement Techniques. The data and approach are novel and appear of good quality. The manuscript could benefit from some attention to detail and some mostly minor clarifications.

A fundamental question that I do not see mentioned or addressed is that ATR spec-

troscopy is highly sensitive to the deposit structure and depth homogeneity. I am missing a clear discussion on how the FG approach compares with PM mass? i.e. if the data gets less consistent with TOR when there is more deposit on the filter? Or not? This seems a critical thing to discuss?

A second related issue is what happens when diurnally different sources are important? E.g. vehicles AM and wood burning PM as the corresponding molecules are now in different layers and as stated above ATR is sensitive to penetration depth with higher sensitivity to molecules closer to the surface. Please comment? Could Phoenix data show something there?

Phoenix is being highlighted in the manuscript, which surprised me as I thought of IMPROVE being a rural network. In any case, even more surprising to me is the Phoenix woodburning. Can you please provide peer reviewed literature that actually supports that biomass burning is so dominant over the whole wintertime (3 months, if they have even 3 months of winter in Phoenix) and all this is not just some mass/deposit heterogeneity artifact as questioned above related to very strong inversion periods?

The manuscript could benefit from more attention to detail. Examples: L14 and throughout the manuscript-Please use subscripts in your molecule numbers. L14 COOH would be carboxylic acid especially when contrasted with carboxylate Figure 3: One molecule (dione) has carbons with 5 bonds (get rid of that double bond in the keto containing cycle). Also for all these species provide the correct names.. not only the random MCM naming. The second molecule is poorly cropped and cut.

Other comments: L19 R2 values are meaningless without n? or discussion of statistical relevance., What is the "Reconstructed Fine Mass Equation", I assume this is some American network thing? (L34) Blanks: why was only ammonium sulfate used as blanks? Not ammonium nitrate or other materials? Figure 2, could you provide quality parameters on the fits? Figure 5: Could you show a trendline with equation here? To get a quantitative idea of the drift? Figure 9: Probability density function are often given

in units that if the curve gets integrated it equals 1. In this manuscript, it seems never normalized and the density always arbitrary? In Fig 9 on some panels one wonders if all areas under the curve are identical though?

---

## Referee Comment (RC3) · Anonymous Referee #3 · 6 Jan 2020

Bürki et al. present a probabilistic modeling framework for estimating organic carbon concentrations and OM/OC ratios from infrared spectroscopy, and they apply it to infrared spectra of PM2.5 samples from 17 monitoring sites of the IMPROVE network. The presented approach is based on previous developments regarding functional group analysis from infrared spectroscopy and statistical calibration strategies for organic aerosol quantification. Here, the authors apply Bayesian calibration to provide plausible estimates for parameters in the probabilistic model, and they obtain OC

concentrations and OM/OC ratios consistent with other estimation approaches. The presented work is well within the scope of AMT. However, in its present form the manuscript is difficult to follow especially for atmospheric scientists who do not use linear algebra in their day-to-day work. Thus, the manuscript requires major revisions to improve clarity and to focus on the novelty of the approach before publication in AMT.

While I appreciate the rigorous explanations in the appendices and in the supplementary material, some parts of the main text either require additional information or should be moved to the supplementary material in order to focus on the main objective of the manuscript. In particular, the introduction of the probabilistic framework in section 1.2 might be revised with the general reader in mind. Also, the description of the cluster analysis might be shortened, and Figures 4 and 6 might be moved to supplement section S3.

When comparing the FTIR estimates of OM/OC with the reconstructed fine mass (RCFM) regression solved both by ordinary least squares (OLS) regression and error-in-variables (EIV) regression, the extensive comparison of OLS and EIV seems distracting. It may be beneficial to briefly introduce both OLS and EIV in section 3.3 but then restrict the comparison in section 4.4 and Figures 8 and 10 to FTIR and only OLS, or only EIV.

Functionalization by aldehyde, peroxide, aromatic, phenolic, organonitrate, and organosulfate groups is not included in the presented set of calibrations. While it is prudent to prioritize functional groups that are expected to be highly abundant, one has to be very careful when interpreting the results, as stated for example when discussing low mass recovery fractions that "we cannot rule out the need to examine additional FGs" (l.349). For example, organosulfates may become more important in situations when biogenic VOCs are processed in anthropogenically influenced air masses. Thus, changes in OM/OC ratios between 2011 and 2013 observed in the RCFM regression but not observed in the FTIR estimates could also indicate an increasing influence of FGs not taken into account in the presented set of calibrations. I recommend a brief

discussion.

Minor comments: Figure 2: What is the reasoning for fitting Weibull distributions to the prior "fractional carbon" coefficients but a normal distribution to the mass recovery fraction? Figure 7 is introduced after Figure 4 and before Figures 5 and 6. Please re-order figure numbers. Conclusions: The final notion that additional constraints from additional measurements such as NMR or photometry can be added is really helpful and important.

Technical comments: Please introduce all abbreviations when first used, e.g. "RCFM" in line 141, L-BFGS-B in l.562, etc. l.5: "For instance, a subset of model..." - please revise the sentence. l.14: Carboxylic acid should be "COOH". l.82 "... proposed an extension to this approach..." l.138/Figure 1: Please indicate the four regions SW/NW/SE/NE by adding lines on the map in Fig. 1. l.165: "L2 norm" might need explanation. l.201: Revise "...regression of to y". l.355: "but it more likely reflects" instead of "but is more likely reflects" l.394: Add parentheses to references. l.451: Remove the extra "from" from "...that estimate OM/OC from from mass balance...". Figure 2: "representing" instead of "representating"

---

## Author Comment (AC1) · 5 Feb 2020

The comment was uploaded in the form of a supplement:
https://www.atmos-meas-tech-discuss.net/amt-2019-333/amt-2019-333-AC1-supplement.pdf

---

## Author Comment (AC2) · 5 Feb 2020

**Response to reviewer comments for manuscript: "Analysis of functional groups in atmospheric aerosols by infrared spectroscopy: method development for probabilistic modeling of organic carbon and organic matter concentrations"**

**Reviewer 1**

The manuscript describes a statistical model and a probabilistic framework to characterize combinations of organic matter, organic carbon and functional groups obtained from the Fourier transform infrared (FTIR) spectra of fine particulate matter (PM2.5). The model was found to be consistent with field measurements of organic carbon (OC). The Development of these models and frameworks is important and timely as they can be used in developing machine learning algorithms.

We thank the reviewer for the encouraging assessment.

1. The abstract is too long. Consider shortening.

    We have substantially shortened the abstract to highlight the essential points of the manuscript.

2. The introduction is missing a discussion on the results obtained from NMR spectroscopy in the characterizing organic aerosols and using multivariate analysis to correlate functional groups with sources of aerosols. For example, see Chapter Two - NMR Studies of Organic Aerosols, Annual Reports on NMR Spectroscopy, Volume 92, 2017, Pages 83-135.

    We thank the reviewer for pointing out this useful review. In current form it is difficult to relate our results to that reported by NMR studies given several unknowns: i) systematic differences in composition between the water soluble fraction (targeted by NMR) and $PM_{2.5}$ analyzed by FTIR in this work, ii) sampling artifacts related to high and low mass loadings (e.g., gas/particle partitioning), and iii) the different scales of functional group ratios used by the two techniques. We hope to dedicate a future study (or series of studies) relating the two techniques for source identification or source apportionment by receptor modeling.

    We have included a citation to the work to indicate its relevance for future comparisons.

3. Figure 4: this figure has 12 sub figures; each is labeled with a 'cluster' number and contain a number of overlapping spectra as judged by the gradation in the line colors. The caption needs to be modified to refer the reader to appropriate table or graph containing the definition of each cluster.

    We have now specified in the caption: "The clustering procedure and interpretation are described in Sections 3.1 and 4.2, respectively."

4. Figure 7: The caption needs to be modified to refer the reader to appropriate table of graph containing the definition of parameters in the y- and x-axes.

    We have now specified in the caption:" "Density" refers to the probability or mass density and the variables are described in Sections 1.1 and 3.2."

5. Figure 8, 9, 10, 11: the use of three-letter abbreviation for seasons that do not correspond to the actual name of the season is confusing: DJF = winter, MAM = spring, JJA = summer, and SON = fall. It is better to use WR = winter, SG = spring, SR = summer, and FL = fall.

We thank the reviewer for the suggestion, but DFJ, MAM, JJA, SON is a standard convention for denoting seasons — we have now written out the name of the months that they stand for in the caption of Figure 8: "X-axes denote seasons: DJF (December, January, February) = winter, MAM (March, April, May) = spring, JJA (June, July, August) = summer, and SON (September, October, November) = fall."

**Reviewer 2**

The manuscript describes a novel approach using FT-IR and functional groups to obtain organic matter to organic carbon ratios. The manuscript fits the scope of Atmospheric Measurement Techniques. The data and approach are novel and appear of good quality. The manuscript could benefit from some attention to detail and some mostly minor clarifications.

We thank the reviewer for the encouraging assessment.

1. *Regarding the FTIR technique*:

- A fundamental question that I do not see mentioned or addressed is that ATR spectroscopy is highly sensitive to the deposit structure and depth homogeneity. I am missing a clear discussion on how the FG approach compares with PM mass? i.e. if the data gets less consistent with TOR when there is more deposit on the filter? Or not? This seems a critical thing to discuss?

- A second related issue is what happens when diurnally different sources are important? E.g. vehicles AM and wood burning PM as the corresponding molecules are now in different layers and as stated above ATR is sensitive to penetration depth with higher sensitivity to molecules closer to the surface. Please comment? Could Phoenix data show something there?

We regret having omitted the details of the analysis that has caused understandable confusion.

One of the coauthors previously used ATR-FTIR to analyze PTFE filters (Coury and Dillner, 2008), which is indeed sensitive to penetration depth and particle morphology. Some of these influences on apparent absorbances of particle samples are described by other publications, including ours (Kortüm, 1969; Harrick, 1979; Milosevic, 2012; Arangio et al., 2019).

For this work, we have used transmission mode analysis as described by Maria et al. (2003) and Ruthenburg et al. (2014). The samples collected in this way are optically thin and so penetration depth is not an issue (transmittance does not reach 0%) (Maria et al., 2003), which is a common concern with transmission mode analysis. Diurnal layering of different particle types can lead to a nonuniform film structure, but empirically we find that the measured transmittance is within measurement error (Debus et al., 2019) regardless of which face of the filter is directed toward the incident beam. We therefore assume that this does not have a detectable impact on our interpretations.

While a "film" of submicron particles analyzed by transmission mode can be modeled as a substance with a single, effective refractive index (Fischer, 1975; Choy, 2016), presence of larger particles can potentially lead to i) scattering and ii) anomalous dispersion (Leisner and Wagner, 2010). The former manifests itself in reduced transmittance and therefore increased apparent absorbance (e.g., Dazzi et al., 2013). This is handled by baseline correction, which removes the scattering contribution from both the PTFE substrate and also the particles

(Takahama et al., 2019). Samples with anomalous dispersion are found in the "anomalous" clusters in this work (referring the reader to Reggente et al., 2019, in which further details are provided), and caveats are provided for their interpretation. Furthermore, the additive mixing model of Bougher-Lambert-Beer (BLB) assumes dilute, non-interacting mixtures, which may not strictly apply in our samples. However, the "inverse calibration" approach of eq. 2 is robust with respect to a certain degree of nonlinearity in the data (Griffiths and Haseth, 2007). These issues would effect the calibration which estimates the molar density $n$ (eq. 2), but not for the overall framework of the FG-OC model (eqs. 1 and 5). The FG-OC model is, in principle, agnostic with respect to the method by which $n$ is estimated.

A separate manuscript describing particle models for mid-infrared analysis has been in preparation and the topics mentioned above will be explored in more depth. For this manuscript, we have added the following statement in Section 2:

"PTFE of ambient and laboratory samples were analyzed nondestructively by FTIR in transmission mode (Maria et al., 2003) after placing them in a custom minichamber purged with air passed through a molecular sieve to remove water vapor and carbon dioxide (Ruthenburg et al., 2014; Debus et al., 2019). Spectra were truncated to the region above 1500 cm$^{-1}$ and baseline corrected (Kuzmiakova et al., 2016) to reduce scattering contributions from the PTFE filter (McClenny et al., 1985) and particles (Takahama et al., 2019). Further details on the sample collection, analysis, and spectra processing steps are described by previous works (Ruthenburg et al., 2014; Reggente et al., 2016; Debus et al., 2019; Takahama et al., 2019)."

2. Phoenix is being highlighted in the manuscript, which surprised me as I thought of IMPROVE being a rural network. In any case, even more surprising to me is the Phoenix woodburning. Can you please provide peer reviewed literature that actually supports that biomass burning is so dominant over the whole wintertime (3 months, if they have even 3 months of winter in Phoenix) and all this is not just some mass/deposit heterogeneity artifact as questioned above related to very strong inversion periods?

We thank the reviewer for pointing out the absence of proper citations which we have now added to Section 4.2. IMPROVE does contain a few urban sites; wood burning in Phoenix, AZ, has previously reported by Ramadan et al. (2011) and the "official fireplace season" lasts through October through February (Pope et al., 2017). In Cluster 9b, 22 out of the 26 samples fall in this period, with the remaining samples coming from September (1 sample) and March (3 samples).

3. The manuscript could benefit from more attention to detail. Examples: L14 and throughout the manuscript-Please use subscripts in your molecule numbers. L14 COOH would be carboxylic acid especially when contrasted with carboxylate.

"carboxylic COO" should have been written "carboxylic COOH" on L14 and we apologize for this error. But in general, there is no molecule number because we use the approach of calibrating to functional groups rather than individual molecules (Anderson and Seyfried, 1948; Allen et al., 1994; Russell, 2003). We have now made it clear in the abstract that we are using functional group calibrations.

4. Figure 3: One molecule (dione) has carbons with 5 bonds (get rid of that double bond in the keto containing cycle). Also for all these species provide the correct names; not only the random MCM naming. The second molecule is poorly cropped and cut.

We thank the reviewer for pointing out these errors. We have revised the figure and included standard names for all molecules shown.

5. Other comments:

- L19 R2 values are meaningless without $n$? or discussion of statistical relevance.

  We did not intend to imply that the agreement was perfect. We have now stated "the resulting calibrations reproduce TOR OC concentrations with reasonable agreement ($r = 0.96$ for 2474 samples) and provide OM/OC values generally consistent with our current best estimate of ambient OC." We had also mistakenly written the value of Pearson's correlation coefficient ($r$) as the coefficient of determination ($R^2$) for OLS regression so this has also been corrected.

- What is the "Reconstructed Fine Mass Equation", I assume this is some American network thing?

  This is an equation of mass closure defined by the IMPROVE network that expresses mass closure between gravimetric $PM_{2.5}$ mass and its major mass constituents. We have included this statement in Section 3.3:

  "For comparison, we estimate OM/OC as interpreted by coefficients of the RCFM equation (a statement of mass closure) [...]"

- (L34) Blanks: why was only ammonium sulfate used as blanks? Not ammonium nitrate or other materials?

  For a limited set of resources there is always a tradeoff between sampling more organic compounds versus non-organic blanks. With regards to ammonium nitrate specifically, including it in our calibrations would not necessarily introduce additional information as we have only used the region of FTIR spectra above 1500 $cm^{-1}$, where the N-H stretch in the ammonium is already captured by the ammonium sulfate.

  The effect of including ammonium nitrate, water, and other non-organic interferents in PLS calibrations has been explored by Boris et al. (2019), who found that organic functional group abundances could be predicted with similar accuracy but with possible differences in the overall model complexity (i.e., number of latent variables). While such conclusions likely depend on the types and proportion of calibration and blank samples, in our calibration set (as well as those of Boris et al., 2019), organic compounds not containing the functional group being calibrated also serve as additional blanks. Therefore, our calibration models are not deficient in blanks for each functional group, though this is an area that can be further explored. As described in the manuscript ("the framework is described generally such that it can [...] systematically evaluate improvements in calibrations with new standards or FGs"), the Bayesian framework presented here provides a means to incorporate new information and evaluate its value based on how it affects parameter posterior distributions.

- Figure 2, could you provide quality parameters on the fits?

  The chi-square statistic for all fits are greater than 300 (normally 1 is considered a "good fit"). Nonparametric models could be used to achieve more precise representations of the generated distributions, but it is not meant to be overly precise since the distribution-generating processes itself is based on approximations — they are dependent on the molecules available in the database, subsets used for estimation (Appendix C), and so on. Parametric models are therefore used to characterize the main features of these distributions (e.g., centrality, dispersion, and symmetry).

- Figure 5: Could you show a trendline with equation here? To get a quantitative idea of the drift?

  We have added a trendline with fit parameters in the caption.

- Figure 9: Probability density function are often given in units that if the curve gets integrated

it equals 1. In this manuscript, it seems never normalized and the density always arbitrary? In Fig 9 on some panels one wonders if all areas under the curve are identical though?

The probability densities are approximated by kernel density estimation (Hastie et al., 2009) (eq. 1), which should integrate to one. Defining a transformed variable $u = (x - x_j)/b$ such that $dx = -b\,du$, each local kernel $K$ with bandwidth $b$ evaluated along $x_j$ for $j = \{1, 2, \ldots, N\}$ integrates to unity (eq. 2); and the resulting integral of the kernel density estimate of $p$ can also be shown to be unity (eq. 3).

$$p(x) = \frac{1}{Nb} \sum_{j=1}^{N} K\left(\frac{x - x_j}{b}\right) \tag{1}$$

$$\int_{-\infty}^{\infty} K(u)du = 1 \tag{2}$$

$$\int_{-\infty}^{\infty} p(x)dx = 1 \tag{3}$$

Some curves may not strictly integrate to one between the axes limits shown as a few OM/OC ratio values lie above 2.4, but most of differences are likely due to perception of density curves that vary in both height and width. We have added to the caption of Figure 7 (now Figure 5) where kernel density estimates are first used: "Non-parametric densities are approximated by kernel density estimation (Hastie et al., 2009) in figures."

**Reviewer 3**

Bürki et al. present a probabilistic modeling framework for estimating organic carbon concentrations and OM/OC ratios from infrared spectroscopy, and they apply it to infrared spectra of PM2.5 samples from 17 monitoring sites of the IMPROVE network. The presented approach is based on previous developments regarding functional group analysis from infrared spectroscopy and statistical calibration strategies for organic aerosol quantification. Here, the authors apply Bayesian calibration to provide plausible estimates for parameters in the probabilistic model, and they obtain OC concentrations and OM/OC ratios consistent with other estimation approaches. The presented work is well within the scope of AMT. However, in its present form the manuscript is difficult to follow especially for atmospheric scientists who do not use linear algebra in their day-to-day work. Thus, the manuscript requires major revisions to improve clarity and to focus on the novelty of the approach before publication in AMT.

We thank the reviewer for the encouraging evaluation.

1. While I appreciate the rigorous explanations in the appendices and in the supplementary material, some parts of the main text either require additional information or should be moved to the supplementary material in order to focus on the main objective of the manuscript. In particular, the introduction of the probabilistic framework in section 1.2 might be revised with the general reader in mind. Also, the description of the cluster analysis might be shortened, and Figures 4 and 6 might be moved to supplement section S3.

We have substantially revised the Section 1.2 to provide additional explanation for a less specialized audience in mind, and moved the details of the cluster analysis from Section 3.2 to Section S3 of the supplement.

We had originally considered the exact same possibility of moving Figures 4 and 6 to the supplement prior to initial submission, but had included them for the reason that our interpretation of why the FTIR OM/OC values are similar between 2011 and 2013 are based on the spectral profiles and their cluster membership.

2. When comparing the FTIR estimates of OM/OC with the reconstructed fine mass (RCFM) regression solved both by ordinary least squares (OLS) regression and error-in-variables (EIV) regression, the extensive comparison of OLS and EIV seems distracting. It may be beneficial to briefly introduce both OLS and EIV in section 3.3 but then restrict the comparison in section 4.4 and Figures 8 and 10 to FTIR and only OLS, or only EIV.

   We have relegated comparisons to the Supplemental document when estimates are similar (i.e., MCMC and Laplace for FTIR), but estimates between the two RCFM regression methods are different enough that removal of one from the final presentation could misrepresent the challenges of interpreting OM/OC from the RCFM approach.

   To make the discussion more clear, we have relabeled OLS and EIV as RCFM-OLS and RCFM-EIV.

3. Functionalization by aldehyde, peroxide, aromatic, phenolic, organonitrate, and organosulfate groups is not included in the presented set of calibrations. While it is prudent to prioritize functional groups that are expected to be highly abundant, one has to be very careful when interpreting the results, as stated for example when discussing low mass recovery fractions that "we cannot rule out the need to examine additional FGs" (l.349). For example, organosulfates may become more important in situations when biogenic VOCs are processed in anthropogenically influenced air masses. Thus, changes in OM/OC ratios between 2011 and 2013 observed in the RCFM regression but not observed in the FTIR estimates could also indicate an increasing influence of FGs not taken into account in the presented set of calibrations. I recommend a brief discussion.

   This is a very good point. We have modified/added the following discussion to Section 3.4:

   "This comparison may support the need for further evaluation along two directions. One is in interpreting $a_{OC}$ from RCFM regression as a surrogate for the OM/OC ratio (Hand et al., 2019). The other is in understanding the changing contributions of FGs not included in our set of calibrations (that also are excluded from or have negligible influence on the spectral cluster analysis) over this period. For instance, recent studies of trends in the Southeast US suggest that aromatic, organosulfate, organonitrate, and peroxide-containing compounds in OM have declined in response to reduced anthropogenic emissions of volatile organic compounds, $SO_2$, and $NO_x$ (the latter two affecting OM through their influence over aqueous-phase reactions and oxidant levels) over the last decades (Pye et al., 2015; Blanchard et al., 2016; Marais et al., 2017; Carlton et al., 2018; Pye et al., 2019). While most of these trends would contradict the direction of discrepancy in OM/OC trends estimated by RCFM and FTIR, the magnitude of changes in emissions and the response of OM likely differ across sites and years considered in this study."

Minor comments:

- Figure 2: What is the reasoning for fitting Weibull distributions to the prior "fractional carbon" coefficients but a normal distribution to the mass recovery fraction?

  The Weibull distribution permits asymmetry, which was important to capture for the two parameters. This has now been noted in the caption.

- Figure 7 is introduced after Figure 4 and before Figures 5 and 6. Please re-order figure numbers.

  We thank the reviewer for catching this error. It has been corrected. Supplement Sections S1 and S2 have now been switched for the same reason.

- Conclusions: The final notion that additional constraints from additional measurements such as NMR or photometry can be added is really helpful and important.

The original manuscript had referred to them in the statement: "or comparison to additional measurements of FGs (Decesari et al., 2007; Ranney and Ziemann, 2016)" but the two methods are now written out explicitly.

Technical comments:

- Please introduce all abbreviations when first used, e.g. "RCFM" in line 141, L-BFGS-B in l.562, etc.

  We thank the reviewer for pointing this out (and all other corrections below) — we have now introduced these abbreviations in the text.

- l.5: "For instance, a subset of model..." - please revise the sentence.

  We have added: "[...] *but* generate substantially different predictions [...]"

- l.14: Carboxylic acid should be "COOH".

  Corrected.

- l.82 "... proposed an extension to this approach..."

  Added: "to"

- l.138/Figure 1: Please indicate the four regions SW/NW/SE/NE by adding lines on the map in Fig. 1.

  This has been done.

- l.165: "L2 norm" might need explanation.

  Now described as "Euclidean distances from the origin when spectra are represented as vectors"

- l.201: Revise "...regression of to y".

  Removed "of"

- l.355: "but it more likely reflects" instead of "but is more likely reflects"

  Corrected.

- l.394: Add parentheses to references.

  Corrected.

- l.451: Remove the extra "from" from "...that estimate OM/OC from from mass balance...".

  Corrected.

- Figure 2: "representing" instead of "representating"

  Corrected.

**References**

Allen, D. T., Palen, E. J., Haimov, M. I., Hering, S. V., and Young, J. R.: Fourier-transform Infrared-spectroscopy of Aerosol Collected In A Low-pressure Impactor (LPI/FTIR) - Method Development and Field Calibration, Aerosol Science and Technology, 21, 325–342, https://doi.org/10.1080/02786829408959719, 1994.

Anderson, J. A. and Seyfried, W. D.: Determination of Oxygenated and Olefin Compound Types by Infrared Spectroscopy, Analytical Chemistry, 20, 998–1006, https://doi.org/10.1021/ac60023a002, 1948.

Arangio, A., Delval, C., Ruggeri, G., Dudani, N., Yazdani, A., and Takahama, S.: Electrospray Film Deposition for Solvent-Elimination Infrared Spectroscopy, Applied Spectroscopy, 73, 638–652, https://doi.org/10.1177/0003702818821330, 2019.

Blanchard, C. L., Hidy, G. M., Shaw, S., Baumann, K., and Edgerton, E. S.: Effects of emission reductions on organic aerosol in the southeastern United States, Atmos. Chem. Phys., 16, 215–238, https://doi.org/10.5194/acp-16-215-2016, 2016.

Boris, A. J., Takahama, S., Weakley, A. T., Debus, B. M., Fredrickson, C. D., Esparza-Sanchez, M., Burki, C., Reggente, M., Shaw, S. L., Edgerton, E. S., and Dillner, A. M.: Quantifying organic matter and functional groups in particulate matter filter samples from the southeastern United States, part I: Methods, Atmospheric Measurement Techniques Discussions, 2019, 1–39, https://doi.org/10.5194/amt-2019-144, 2019.

Carlton, A. G., de Gouw, J., Jimenez, J. L., Ambrose, J. L., Attwood, A. R., Brown, S., Baker, K. R., Brock, C., Cohen, R. C., Edgerton, S., Farkas, C. M., Farmer, D., Goldstein, A. H., Gratz, L., Guenther, A., Hunt, S., Jaeglé, L., Jaffe, D. A., Mak, J., McClure, C., Nenes, A., Nguyen, T. K., Pierce, J. R., de Sa, S., Selin, N. E., Shah, V., Shaw, S., Shepson, P. B., Song, S., Stutz, J., Surratt, J. D., Turpin, B. J., Warneke, C., Washenfelder, R. A., Wennberg, P. O., and Zhou, X.: Synthesis of the Southeast Atmosphere Studies: Investigating Fundamental Atmospheric Chemistry Questions, Bulletin of the American Meteorological Society, 99, 547–567, https://doi.org/10.1175/BAMS-D-16-0048.1, 2018.

Choy, T. C.: Effective medium theory: principles and applications, Oxford University Press, 2nd edn., https://doi.org/10.1093/acprof:oso/9780198705093.001.0001, 2016.

Coury, C. and Dillner, A. M.: A method to quantify organic functional groups and inorganic compounds in ambient aerosols using attenuated total reflectance FTIR spectroscopy and multivariate chemometric techniques, Atmospheric Environment, 42, 5923–5932, https://doi.org/10.1016/j.atmosenv.2008.03.026, 2008.

Dazzi, A., Deniset-Besseau, A., and Lasch, P.: Minimising contributions from scattering in infrared spectra by means of an integrating sphere, Analyst, 138, 4191–4201, https://doi.org/10.1039/C3AN00381G, 2013.

Debus, B., Takahama, S., Weakley, A. T., Seibert, K., and Dillner, A. M.: Long-Term Strategy for Assessing Carbonaceous Particulate Matter Concentrations from Multiple Fourier Transform Infrared (FT-IR) Instruments: Influence of Spectral Dissimilarities on Multivariate Calibration Performance, Applied Spectroscopy, 73, 271–283, https://doi.org/10.1177/0003702818804574, 2019.

Fischer, K.: Mass absorption indices of various types of natural aerosol particles in the infrared, Applied Optics, 14, 2851–2856, https://doi.org/10.1364/AO.14.002851, 1975.

Griffiths, P. and Haseth, J. A. D.: Fourier Transform Infrared Spectrometry, John Wiley & Sons, In, 2[nd] edn., 2007.

Hand, J., Prenni, A., Schichtel, B., Malm, W., and Chow, J.: Trends in remote PM2.5 residual mass across the United States: Implications for aerosol mass reconstruction in the IMPROVE network, Atmospheric Environment, 203, 141 – 152, https://doi.org/10.1016/j.atmosenv.2019.01.049, 2019.

Harrick, N. J.: Internal Reflection Spectroscopy, Harrick Scientific Corp., 1979.

Hastie, T., Tibshirani, R., and Friedman, J.: The elements of statistical learning: data mining, inference, and prediction, Springer Verlag, 2009.

Kortüm, G.: Reflectance Spectroscopy: Principles, Methods, Applications, Springer, 1969.

Kuzmiakova, A., Dillner, A. M., and Takahama, S.: An automated baseline correction protocol for infrared spectra of atmospheric aerosols collected on polytetrafluoroethylene (Teflon) filters, Atmospheric Measurement Techniques, 9, 2615–2631, https://doi.org/10.5194/amt-9-2615-2016, 2016.

Leisner, T. and Wagner, R.: Infrared Spectroscopy of Aerosol Particles, chap. 1, pp. 3–24, CRC Press, 2010.

Marais, E. A., Jacob, D. J., Turner, J. R., and Mickley, L. J.: Evidence of 1991–2013 decrease of biogenic secondary organic aerosol in response to $SO_2$ emission controls, Environmental Research Letters, 12, 054 018, https://doi.org/10.1088/1748-9326/aa69c8, 2017.

Maria, S. F., Russell, L. M., Turpin, B. J., Porcja, R. J., Campos, T. L., Weber, R. J., and Huebert, B. J.: Source signatures of carbon monoxide and organic functional groups in Asian Pacific Regional Aerosol Characterization Experiment (ACE-Asia) submicron aerosol types, Journal of Geophysical Research-Atmospheres, 108, https://doi.org/10.1029/2003JD003703, 2003.

McClenny, W. A., Childers, J. W., Rōhl, R., and Palmer, R. A.: FTIR transmission spectrometry for the nondestructive determination of ammonium and sulfate in ambient aerosols collected on teflon filters, Atmospheric Environment, 19, 1891–1898, https://doi.org/10.1016/0004-6981(85)90014-9, 1985.

Milosevic, M.: Internal Reflection and ATR Spectroscopy, Chemical Analysis: A Series of Monographs on Analytical Chemistry and Its Applications, John Wiley & Sons, Inc., 2012.

Pope, R., Stanley, K. M., Domsky, I., Yip, F., Nohre, L., and Mirabelli, M. C.: The relationship of high PM2.5 days and subsequent asthma-related hospital encounters during the fireplace season in Phoenix, AZ, 2008–2012, Air Quality, Atmosphere & Health, 10, 161–169, https://doi.org/10.1007/s11869-016-0431-2, 2017.

Pye, H. O. T., Luecken, D. J., Xu, L., Boyd, C. M., Ng, N. L., Baker, K. R., Ayres, B. R., Bash, J. O., Baumann, K., Carter, W. P. L., Edgerton, E., Fry, J. L., Hutzell, W. T., Schwede, D. B., and Shepson, P. B.: Modeling the Current and Future Roles of Particulate Organic Nitrates in the Southeastern United States, Environmental Science & Technology, 49, 14 195–14 203, https://doi.org/10.1021/acs.est.5b03738, 2015.

Pye, H. O. T., D'Ambro, E. L., Lee, B. H., Schobesberger, S., Takeuchi, M., Zhao, Y., Lopez-Hilfiker, F., Liu, J., Shilling, J. E., Xing, J., Mathur, R., Middlebrook, A. M., Liao, J., Welti, A., Graus, M., Warneke, C., Gouw, J. A. d., Holloway, J. S., Ryerson, T. B., Pollack, I. B., and Thornton, J. A.: Anthropogenic enhancements to production of highly oxygenated molecules from autoxidation, Proceedings of the National Academy of Sciences, 116, 6641–6646, https://doi.org/10.1073/pnas.1810774116, 2019.

Ramadan, Z., Song, X.-H., and Hopke, P. K.: Identification of Sources of Phoenix Aerosol by Positive Matrix Factorization, Journal of the Air & Waste Management Association, 2011.

Reggente, M., Dillner, A. M., and Takahama, S.: Predicting ambient aerosol thermal-optical reflectance (TOR) measurements from infrared spectra: extending the predictions to different years and different sites, Atmospheric Measurement Techniques, 9, 441–454, https://doi.org/10.5194/amt-9-441-2016, 2016.

Reggente, M., Dillner, A. M., and Takahama, S.: Analysis of functional groups in atmospheric aerosols by infrared spectroscopy: systematic intercomparison of calibration methods for US measurement network samples, Atmospheric Measurement Techniques, 12, 2287–2312, https://doi.org/10.5194/amt-12-2287-2019, 2019.

Russell, L. M.: Aerosol organic-mass-to-organic-carbon ratio measurements, Environmental Science & Technology, 37, 2982–2987, https://doi.org/10.1021/es026123w, 2003.

Ruthenburg, T. C., Perlin, P. C., Liu, V., McDade, C. E., and Dillner, A. M.: Determination of organic matter and organic matter to organic carbon ratios by infrared spectroscopy with application to selected sites in the IMPROVE network, Atmospheric Environment, 86, 47–57, https://doi.org/10.1016/j.atmosenv.2013.12.034, 2014.

Takahama, S., Dillner, A. M., Weakley, A. T., Reggente, M., Bürki, C., Lbadaoui-Darvas, M., Debus, B., Kuzmiakova, A., and Wexler, A. S.: Atmospheric particulate matter characterization by Fourier transform infrared spectroscopy: a review of statistical calibration strategies for carbonaceous aerosol quantification in US measurement networks, Atmospheric Measurement Techniques, 12, 525–567, https://doi.org/10.5194/amt-12-525-2019, 2019.